# Improving the representation of anthropogenic $CO_2$ emissions in climate models:
# impact of a new parameterization for the Community Earth System Model (CESM)

Andrés Navarro[1], Raúl Moreno[1] and Francisco J. Tapiador[1]

[1]Institute of Environmental Sciences (ICAM), University of Castilla-La Mancha, Toledo, 45004, Spain.

*Correspondence to*: A. Navarro (Andres.Navarro@uclm.es)

**Abstract.** ESMs (Earth System Models) are important tools that help scientists understand the complexities of the Earth's climate. Advances in computing power have permitted the development of increasingly complex ESMs and the introduction
of better, more accurate parameterizations of processes that are too complex to be described in detail. One of the least well-controlled parameterizations involves human activities and their direct impact at local and regional scales. In order to improve the direct representation of human activities and climate, we have developed a simple, scalable approach that we have named the POPEM module (POpulation Parameterization for Earth Models). This module computes monthly fossil fuel emissions at grid point scale using the modeled population projections. This paper shows how integrating POPEM
parameterization into the CESM (Community Earth System Model) enhances the realism of global climate modeling, improving this beyond simpler approaches. The results show that it is indeed advantageous to model $CO_2$ emissions and pollutants directly at model grid points rather than using the same mean value globally. A major bonus of this approach is the increased capacity to understand the potential effects of localized pollutant emissions on long-term global climate statistics, thus assisting adaptation and mitigation policies.

## 1 Introduction

The Earth system is a complex interplay of physical, chemical and biological processes that interact in non-linear ways (Ladyman et al., 2013; Lorenz, 1963; Rind, 1999; Williams, 2005). Much effort has been devoted to understanding these complex interactions, and several improvements have been made since the end of the last century.

One of the most important advances in this field has been the use of coupled numerical climate models, dubbed Earth System Models, or ESMs (Edwards, 2011; Flato, 2011; Schellnhuber, 1999). These models aim to simulate the complex interactions of the atmosphere, ocean, land surface, and cryosphere, together with the carbon and nitrogen cycles (Giorgetta et al., 2013; Hurrell et al., 2013; Martin et al., 2011; Schmidt et al., 2014).

However powerful, climate models are far from being perfect (Hargreaves, 2010; Hargreaves and Annan, 2014). Unresolved processes (Williams, 2005), limited computational resources (Shukla et al., 2010; Washington et al., 2009), and model uncertainties (Baumberger et al., 2017; Lahsen, 2005; Steven and Bony, 2013) are ongoing issues that still require attention and further improvement.

One of the fields most in need of development is the inclusion of co-evolutionary dynamical interactions of the socioeconomic dimension into global models with other Earth system components (Nobre et al., 2010; Robinson et al., 2017; Sarofim and Reilly, 2011). Human activity was a major driver of change in the Earth System in the recent past (Alter et al., 2017; Barnett et al., 2008; Crutzen, 2002), and it now dominates the natural system (Ruth, et al. 2011). However, most global models use basic socioeconomic assumptions about the behavior of societies and are only unidirectionally linked to the biogeophysical part of the Earth system (Müller-Hansen et al., 2017; Smith et al., 2014). The standard way of introducing anthropogenic climate change into ESMs is through Representative Concentration Pathways (RCPs). These are consistent sets of projections involving only radiative forcing components (van Vuuren et al., 2011), but which represent a step forward from the scenario approach of the last decade (Moss et al., 2010; van Vuuren et al., 2014; van Vuuren and Carter, 2014). However, RCPs are not fully-integrated socioeconomic parameterizations but rather estimates for describing plausible trajectories of human climate change drivers (Moss et al., 2010; Vuuren et al., 2012). They provide simplified accounts of human activities and processes from one-way coupled Integrated Assessment Models (IAMs, Müller-Hansen et al., 2017).

The use of RCPs is advantageous because they provide a set of pathways that serve to initialize climate models. However, two major problems remain within this approach. Firstly, human activities are not intrinsically embedded into the ESM, impeding sensitivity studies. Secondly, because of the weak coupling of IAMs, they cannot capture the sometimes counterintuitive bidirectional feedback and nonlinearity between the socioeconomic and natural subsystems (Motesharrei et al. 2016; Ruth et al. 2011). Good examples that illustrate the importance of including such bidirectional feedbacks feature in the HANDY model (Motesharrei et al. 2014) which has been used to analyze the key mechanisms behind societal collapses using the predator-prey model.

The RCP approach has been used in climate models because of its low computational cost. However, advances in computational resources now allow to parameterize human-Earth processes in a more detailed way, including the inclusion of population dynamics into the modeling, as in the POPEM (POpulation Parameterization for Earth Models) module (Navarro et al., 2017).

One important, but sometimes overlooked process is the direct, regional effect of anthropogenic greenhouse gas (GHG) emissions. Although some GHGs quickly mix in the atmosphere (IPCC, 2014a), their mixing times and lifetimes vary

(Archer et al., 2009; Prather, 2007), and localized emissions may produce a transient response in the atmosphere. Given the highly non-linear character of the processes involved, it is not unreasonable to assume that accounting for geographical variability is significant, and the spatial and time distribution of these emissions may affect global climate (Alter et al., 2017; Grandey et al., 2016; Guo et al., 2013). This hypothesis has seldom been investigated, as most current models treat certain GHG emissions as a homogeneously distributed forcing. Thus, for instance, the most typical CESM (Community Earth System Model) simulations prescribe a $CO_2$ concentration on the assumption that it is well-mixed in the atmosphere (Neale et al., 2012).

This paper describes the results of a 50-year simulation with a simple parameterization of fossil fuel $CO_2$ emissions at model grid point scale, integrating the POPEM module into the CESM. The aim of this paper is to show that this grid point scale modeling of anthropogenic $CO_2$ emissions (and other pollutants) represents an improvement over simpler approaches, and leads to better representation of the geographical variability of precipitation.

The purpose of the new modeling is not only to improve precipitation and temperature estimates, but also help understand the carbon cycle feedback, and evaluate the climate sensitivity of the Earth under alternative GHG emission scenarios. While our focus here is anthropogenic $CO_2$ emissions, the POPEM parameterization can accommodate other GHGs and human-dependent processes in order to advance CESMs towards a comprehensive, fully-coupled modeling of anthropogenic dynamics in the global climate.

The paper is organized as follows: in section 2, we present the validation of the POPEM standalone mode and set the framework for evaluating the impact of POPEM parameterization –its incorporation into the CESM and the testing framework; in section 3, we compare the outputs of CONTROL and POPEM runs and see how they compare with observations. In the conclusion and future work section, we highlight the importance of the dynamical modeling of anthropogenic emissions at grid point scale to better represent the socioeconomic parameters in the CESM model and improve precipitation estimates.

## 2. Material and methods

### 2.1 The CESM model

The Community Earth System Model (CESM) is a state-of-the-art ESM and probably the most widely used climate model. It was developed and is maintained by the National Center for Atmospheric Research (NCAR), with contributions from external researchers funded by the U.S. Department of Energy (DOE), the National Aeronautics and Space Administration (NASA), and the National Science Foundation (NSF) (Hurrell et al., 2013). CESM is an ESM comprising a system of multi-geophysical components, which periodically exchange two-dimensional boundary data in the coupler (Craig et al., 2012). It

consists of five component models and one central coupler component: the atmosphere model CAM (Community Atmosphere Model; (Tilmes et al., 2015), the ocean model POP (Parallel Ocean Program; (Kerbyson and Jones, 2005); the land model CLM (Community Land Model; (Lawrence et al., 2011); the sea ice model CICE (Community Ice Code; (Hunke and Lipscomb, 2008); and the ice sheet model CISM (Community Ice Sheet Model; (Lipscomb et al., 2013).

CESM –formerly the Community Climate System Model (CCSM)- was conceived as a coupled atmospheric-oceanic circulation model (Boville and Gent, 1998; Collins et al., 2006; Gent et al., 2011; Hurrell et al., 2013; Williamson, 1983). Since the release of the first version, CESM has evolved into a complex Earth System Model now used in different fields. This includes research into atmospheric (Bacmeister et al., 2014; Liu et al., 2012; Yuan et al., 2013), biogeochemical

(Lehner et al., 2015; Nevison et al., 2016; Val Martin et al., 2014), and human-induced processes (Huang and Ullrich, 2016; Levis et al., 2012; Oleson et al., 2011), as well as others. The core code of CESM has also been utilized by various research centers for developing their own models (norESM, Bentsen, 2013; CMCC–CESM–NEMO, Fogli and Iovino, 2014; MIT IGSM-CAM, Monier et al., 2013). CESM has been used in many hundreds of peer-reviewed studies to better understand climate variability and climate change (Hurrell et al., 2013; Kay et al., 2015; Sanderson et al., 2017). Simulations performed

with CESM have made a significant contribution to international assessments of climate, including those of the Intergovernmental Panel on Climate Change (IPCC) and the CMIP5/6 project (Coupled Model Intercomparison Project Phase 5/6) (Eyring et al., 2016; IPCC, 2014b; Taylor et al., 2012).

A major advantage of CESM over other ESMs is its availability. Some climate models are developed by scientific groups

and access to the source code is limited. The CESM source code is free and available to download from the NCAR website. This approach helps improve the model by setting up a framework for collaborative research and makes the model fully auditable. CESM is a good example of a 'full confidence level' model, after Tapiador et al. (2017), where many 'avatars' of the code are routinely run in several independent research centers, and there is an entire community improving the model and reporting on issues and results. However, the model is not immune to bias. One important shortcoming is the poor

representation of precipitation in terms of spatial structure, intensity, duration, and frequency (Dai, 2006; Tapiador et al., 2018; Trenberth et al., 2017, Trenberth et al., 2015). Another major bias is the anomalous warm surface temperature in coastal upwelling regions (Davey et al., 2001; Justin Small, 2015; Richter, 2015).

## 2.2 POPEM specifics and standalone validation

### 2.2.1 POPEM parameterization model overview

The POPEM module is a demographic projection model coded in FORTRAN that is intended to estimate monthly fossil fuel $CO_2$ emissions at model grid point scale using population as the input. Due to a lack of actual GHG measurements at

appropriate spatial and temporal scales, it is necessary to use a proxy. For this, POPEM employs population, the evolution of which is modeled using external parameters that feed the module. The idea of using population as proxy is not new, and population density has previously been used to downscale national $CO_2$ emissions (Andres et al., 1996, 2016). However, these inventories were not dynamical, but instead tied to historical data so it is not possible to use them either to estimate future changes in emissions, or coupled with other components of the model. This change represents an important advance in the way emissions are computed. Thus, POPEM uses a bottom-up approach, where emissions are calculated at cell level on the basis of population dynamics, while global inventories use a top-down approach, which is less flexible when coupled with other components of the ESM.

The demographic/emissions module presented here is an updated version of the demographic module explained in Navarro et. al (2017). The differences between the versions are minimal. They involve better approximation of emissions in highly polluting regions with poor population data, such as China; a better estimate for coastal zones and country limits; and a change in the model time step for more efficient coupling with CESM. The inclusion of these changes results in more accurate emissions estimates when compared with inventories than the previous version did. However, the model is not immune to bias. The most important limit is the degradation of the model outputs when there is increased spatial resolution – resolution of 0.25$^{o}$ and higher–.

Detailed information on POPEM and its validation in the demographic realm can be found in (Navarro et al., 2017). In short, from an initial condition, the routine computes the population for each model grid point in a 2D matrix and then calculates fossil fuel $CO_2$ emissions using per capita emission rates by nations. The process is repeated for each time step (e.g. annually) throughout the simulation period.

Figure 1 about here

As seen in Figure 1, POPEM stores gridded emission data in a 3D array (time, latitude and longitude) to be used by the modified version of the *co2_cycle* module. This module reads emissions data and passes this to the *atm_comp_mct,* which calculates the total amount of $CO_2$ emissions from different sources (land, ocean and fossil fuel).

**2.2.2 POPEM trend verification**

Prior to coupling POPEM with CESM we performed several tests to evaluate its ability to reproduce historical population trends and $CO_2$ emissions. To do this, we ran the module in standalone mode. In a first test, we ran a short simulation (1950-2013) and compared the emissions data with a standard emissions inventory (CDIAC). In a second test, POPEM was run for

70 years (1950-2020) and population estimates were validated against the UN (United Nations) population statistics database for those years when data was available.

As shown in Figure 2, POPEM is capable of satisfactorily simulating the dynamics of the population. Comparison with UN data shows good agreement. However, POPEM presents slight differences from the reference data in some regions. Several of these discrepancies can be explained by the initial model conditions; POPEM uses the same age distribution inside each grid cell to initiate the model (only for the first time-step). This distribution is based on the global average age structure.

Consequently, the model overestimates the population in those regions with a more elderly age structure, i.e., Europe and North America, and underestimates areas with younger populations, i.e., Latin America and Asia.

These disparities in population counts have a diverse effect on the outputs in terms of GHG emissions. Thus, for example, the bias in Europe seems to be more important than the bias in Latin America and Oceania. Two principal reasons could

explain this: population size, as Europe has a larger population than Oceania, so there is greater bias in the $CO_2$ emissions estimation; and the per capita emissions rate, as Latin American countries have lower per capita emissions rates than European nations.

It is worth noting here that the POPEM outputs in Figure 2 are clearly non-linear and thus not trivially derived from simply

extrapolating population. The North American estimate of $CO_2$ emissions (second row from the bottom) clearly shows the added value introduced by the model.

Figure 3 shows how POPEM distributes $CO_2$ emissions for different years in the recent past. In 1950, the majority of emissions tended to be concentrated in the USA and Europe, while in 2000, China, the USA and India were the most

polluting countries. This is consistent with the literature: POPEM's estimates generally agree with the emissions maps for the recent past (Andres et al., 1996; Boden et al., 2017; Oda et al., 2018; Rayner et al., 2010), as well as with regional studies on $CO_2$ emissions (Gately et al., 2013; Gurney et al., 2009).

The regionalized distribution of emissions depicted in Figure 3 represents a vast improvement over the standard procedure of using globally-averaged emissions. Even accounting for rapid mixing of GHGs gases, transient effects are likely to appear given the hemispheric contrast and regional differences in the emissions. The differences in Asia are illustrative of the economic changes in the recent past and the exponential pace of industrialization in that region.

## 2.3 CESM experimental setup

The CESM used in this work is based on version 1.2.2 (http://www.cesm.ucar.edu/models/). This set includes active components for the atmosphere, land, ocean, and sea ice, all coupled by a flux coupler. The latest atmospheric module CAM5 (Neale et al., 2012) is used to introduce more accurate modeling of atmospheric physics. Additionally, the carbon cycle module is included in CESM's atmosphere, land, and ocean components (Lindsay et al., 2014).

We ran an experiment at 1.9$^o$ degrees of spatial resolution for the period 1950-2000. Two simulations were performed to analyze the effects of the regionalized emissions (Figure 3) on the CESM. Our control case used homogeneous $CO_2$ concentration parameters (standard procedure in ESMs), while the POPEM case used geographically-distributed $CO_2$ emissions data. In the latter, the POPEM module was coupled with the atmospheric $CO_2$ flux routine to provide monthly gridded $CO_2$ emissions. The gridded data was used at each time step by the atmospheric routine. Apart from this change, both simulations were identical in order to identify the effects (if any) of the POPEM parameterization.

## 2.4 Validation data

### 2.4.1 GPCP data set

Precipitation is one of the key elements for balancing the energy budget, and one of the most challenging aspects of climate modeling. Hence, high quality estimates of precipitation distribution, amount and intensity are essential (Hou et al., 2014; Kidd et al., 2017; Xie and Arkin, 1997). While there are many sources of precipitation data to be used as a reference (see (Tapiador et al., 2012) for a review), only a few qualify as 'full confidence level validation data' (Tapiador et al., 2017).

The Global Precipitation Climatology Project (GPCP; Adler et al., 2016) has several products suitable for validating climate models. GPCP-Monthly is one of the most popular precipitation data sets for climate variability studies. It combines data from rain gauge stations and satellite observations to estimate monthly rainfall on a 2.5-degree global grid from 1979 to the present. The careful combination of satellite-based rainfall estimates results in the most complete analysis of rainfall available to date over the global oceans, and adds necessary spatial detail to rainfall analyses over land. Due to its relevance and global coverage, it has been widely used for validating precipitation in climate models (Li and Xie, 2014; Pincus et al., 2008; Stanfield et al., 2016; Tapiador, 2010).

### 2.4.2 CRU data set

Global surface temperature data sets are an essential resource for monitoring and understanding climate variability and climate change. One of the most commonly used data sets is produced by The Climate Research Unit at the University of East Anglia (CRU). This group produces a high-resolution gridded climate dataset for land-only areas, the Climate Research Unit Timeseries (CRUTS; Harris et al., 2014). CRUTS contains monthly time series of ten climate variables, including surface temperature. The data set is derived from monthly observations at meteorological stations. Station anomalies are interpolated into 0.5º latitude/longitude grid cells covering the global land surface and combined with existing climatology data to obtain absolute monthly values (New et al., 1999, 2000). It is commonly used in the validation of climate models because of its confidence levels, together with temporal and spatial coverage, and the fact it compiles station data from multiple variables from numerous data sources into a consistent format (Christensen and Boberg, 2012; Hao et al., 2013; Liu et al., 2014; Nasrollahi et al., 2015).

### 2.4.3 GISTEMP data set

NASA's GISTEMP (GISS Surface Temperature Analysis) is a global surface temperature change dataset (Hansen and Lebedeff, 1987; see Hansen et al. 2010 for an updated version). It combines land and ocean surface temperatures to create monthly temperature anomalies at $2^o$ x $2^o$ degrees of spatial resolution. The use of anomalies reduces the estimation error in those places with incomplete spatial and temporal coverage (Hansen and Lebedeff, 1987). The anomalies are calculated over a fixed base period (1951-1980) that makes the anomalies consistent over long periods of time.

The first version was originally conceived only for land areas (Hansen and Lebedeff, 1987) but in 1996 marine surface temperatures were added (Hansen et al., 1996). The updated version of GISTEMP includes satellite-observed nightlights to identify stations located in extreme darkness and adjust temperature trends of urban stations for non-climatic factors (Hansen et al. 2010). Just like CRUTS, GISTEMP is commonly used to validate climate models because of its coverage and confidence levels (Baker and Taylor, 2016; Brown et al., 2015; Neely et al., 2016, Peng et al., 2015).

### 3. Results and discussion

### 3.1 Comparison between the CONTROL and POPEM runs

It is worth stressing that a parameterization which performs well when tested for the variable it models does not necessarily translate into an overall improvement of the other variables in the model. An accepted practice in climate modeling is to tune ESMs by adjusting some parameters to achieve a better agreement with observations (Hourdin et al., 2017; Mauritsen et al.,

2012). These adjustments to specific targets may, however, decrease the model's overall performance (Hourdin et al., 2017), and give poor scores for variables other than those tuned. Thus, for example, if a model is biased with respect to aerosol concentrations or humidity, then improved parameterization of cloud formation may worsen the performance of the model with regard to precipitation (Baumberger et al., 2017). This mismatch can be caused by model over-specification, or over-tuning.

The first step in evaluating the new parameterization is to compare the outputs with a control simulation to make sure the new addition does not negatively interact with the dynamical core or spoil the contributions of rest of the parameterizations. Figure 4 shows that this is not case with the POPEM parameterization, which does not negatively affect the outputs of precipitation and temperature. Rather, both variables are now closer to the observed data than they were in the control run, especially in terms of reducing the double ITCZ (Intertropical Convergence Zone), which artificially features in global models (Mechoso et al., 1995; for a recent analysis of double ITCZ in CMIP5 models see Oueslati and Bellon, 2015).

Figure 4 about here

Figure 4A shows that there is just a slight discrepancy in the absolute difference in rainfall between the GPCP and CESM simulations (The first and the third quartiles of the distribution remain between $\pm$ 0.4 mm/day). Grid point to grid point comparison between the model and GPCP indicates the ability of CESM to reproduce the spatial distribution of precipitation. In both simulations, the CESM exhibits a good correlation coefficient (0.72 $R^2$) compared with the reference data (Figure 4C). The results are even better for temperature (0.88 $R^2$; Figure 4D).

Direct comparison of aggregated data is a standard procedure for gauging model abilities. Figure 5 compares two latitude-time graphs for precipitation (A) and surface temperature (B), both for the CONTROL case and for the new POPEM parameterization.

Figure 5 about here

It is clear from Figures 5A and 6A that POPEM does alter the spatial pattern of precipitation and exerts a definite effect on the climate pattern, as the module reduces the otherwise exaggerated ITCZ precipitation in the Southern Hemisphere reported by several authors (Hwang and Frierson, 2013; Lin and Xie 2014).

Disparities in temperature between the CONTROL and POPEM runs are apparent at high latitudes. In this case, POPEM produces lower temperatures at both poles, a result which deserves further attention (Figures 5B and 6B).

Figure 6 about here

There are also important differences in precipitation in the 30N-30S band. Here POPEM reduces model bias, especially in the Southern Hemisphere and on the Tibetan Plateau (see section 3.2 for more details). On the other hand, POPEM departs from the control simulation in the Asia-Pacific region between 10N-10S. This result reinforces the double ITCZ bias in this area.

These results show that the POPEM parameterization generally agrees with historical data for population, and also compares well with the control simulation in the sense of addressing some of the known biases in precipitation and temperature, offering a more detailed version of $CO_2$ emissions at a relatively cheap computational cost. As discussed above, the CONTROL run uses global concentration values to include $CO_2$ on the assumption that it is well-mixed in the atmosphere (Neale et al., 2012). This assumption reduces the computational burden of the simulation but does not allow for precise emissions modeling in the future. This is an important aspect for regionalized emissions scenarios, since even if the new parameterization is not significantly better than the old approach (but no worse), it is desirable as it allows sensitivity analyses, such as evaluating the effects of the U.S. leaving the Paris agreement.

Potential applications of POPEM include not only sensitivity analyses of local $CO_2$ emissions policies, but also the added feature of performing tests for 'what-if' scenarios. One interesting example would be the climate response under the hypothesis that China and India –the most populated countries in the world- reach US $CO_2$ per capita emissions rates. Another 'what-if' scenario would be the climate response of an increasingly urbanized world. In both cases, POPEM provides a flexible framework for testing the alternative hypotheses.

The realism of the ESM will be enhanced with a fully-coupled system. Such a fully-fledged ESM will include bidirectional feedback between POPEM and CESM to evaluate the effects of climate change on population dynamics and emissions.

## 3.2 Validation against observational data sets

Once it has been verified that the new parameterization does not worsen the modeling, the next step in evaluating the performances is comparing the simulation outputs for both the CONTROL run and the POPEM module using actual observational data. Direct comparisons with historical data can help show whether or not a climate model correctly represents the climate of the past. However, although observational measurements are often considered the ground truth to validate models against, it is important to be aware that measurements have their own uncertainties (Tapiador et al. 2017).

Figure 7 shows a comparison of CESM precipitation simulations for the period 1980-2000 using the GPCP. It is apparent that there is an overall consensus, even though there are differences. Despite these known biases, the model agrees with the observations on the major features of global precipitation.

5                                        Figure 7 about here

The improvements in parameterizing emissions become clearer if we focus on specific regions. For the El Niño-4 area, there are statistically-significant differences (at the 0.05 significance level) between both the CONTROL run and the POPEM modeling when compared with the reference data. This observation illustrates the limitations of the modeling and the need of

advances in the parameterizations. However, for this area the correlation ($R^2$) between POPEM and GPCP is slightly better than CONTROL and GPCP (0.706 $R^2$ versus 0.692 $R^2$).

The real added value, however, is not in a better estimation of the totals but in the ability of POPEM to better capture the structure of the precipitation. Figure 8 shows the histograms of mean precipitation in the El Niño-4 area using the POPEM

parameterization (top), the standard forcing approach (CONTROL, middle), and the reference GPCP estimates (bottom). While the CONTROL simulation severely overestimates the low end of the distribution, POPEM gives a more realistic value. This result is not apparent in the otherwise improved correlation of POPEM, and is also buried in the box plots.

El Niño-4 is important because it presents a lower variance in the SST (sea surface temperature) than any other of the El

Niño areas, playing a key role in identifying El Niño Modoki events (Ashok et al., 2007; Ashok and Yamagata, 2009; Yeh et al., 2009). The consequences of such events are severe disruptions in human activities due to the increased risk of droughts, heat waves, poor air quality and wildfires (McPhaden et al., 2006). Thus, precise modeling of the processes in this sector of the Pacific is extremely important.

25                                        Figure 8 about here

Another important benefit of POPEM is the reduction of the double ITCZ bias in the Southern Hemisphere. Although a small change can be inferred from Figure 7A-B, the improvement is buried in the annual mean precipitation maps. Figure 9A shows that the POPEM results are closer to observations of the intra-annual variability of precipitation, especially for the

driest months (June-October).

Figure 9 about here

The figure also shows slight improvements for another two typical biases seen in CESM, namely the excess precipitation in the Tibetan Plateau (Chen and Frauenfeld, 2014; Su et al., 2013; Figure 9C) and the bias in some areas affected by the Asian-Australian monsoon (AAM), such as the Australia Top End (Meehl and Arblaster, 1998; Meehl et al. 2012; Figure 9B).

The results for the El Niño-4 area show that detailed, grid-point emissions of GHG improves the quantification of precipitation in dry areas, in agreement with our hypothesis about the benefits of locally-distributed versus global mean forcings. Also, the double ITCZ example shows that the transient effects of regionalized GHG emissions may even translate into (long) 50-yr climatologies, meaning there is room for improvement in the 'rapidly mixing, well-mixed gases' forcing

approach.

Figure 10 compares the annual mean temperatures for the period 1950-2000. CESM simulations show a significant bias in high latitudes of the Northern Hemisphere (cfr. Figures 10A and 10B). In these areas, the model produces colder temperatures than those registered in the CRUTS reference data but this is also an issue in the CONTROL run. This

deviation is also apparent in Figure 4B, where negative values lie away from the idealized regression line, and indicate further improvement of the CESM.

Figure 10 about here

The bias is also reproduced when compared with temperature anomalies for a specific region. Thus, for instance, CESM gives poor scores in the Barents Sea area (Figure 11; top) while POPEM obtains better results for the Bering Sea, especially in the Russian part (Figure 11; middle). Here, POPEM gives more realistic values for the period 1970-1998 but, even with the improvement, the model still overestimates the temperature anomaly.

Figure 11 about here

If we focus on global temperature anomalies, CESM simulations are able to reproduce the progressive increase in the temperature anomaly (Figure 12; top). However, the CONTROL case simulates a sharp drop at the end of the period (1990-1999), while POPEM portrays this change as smooth, in agreement with the observations.

Figure 12 about here

The differences between CONTROL and POPEM are better demonstrated when comparing land and ocean separately (Figure 12; middle and bottom). While the temperature anomalies for land are quite similar in both cases, POPEM provides a better representation of the ocean tendency from 1992 onwards, and that translates to an overall improvement (Figure 12, top).

### 3.3 Validation against ESPI and ONI indices

The El Niño-Southern Oscillation (ENSO) is the most dominant inter-annual climate variation in the tropics. It occurs when seasonally averaged sea surface temperature anomalies in the eastern Pacific Ocean exceed a given threshold and cause a shift in the atmospheric circulation (Trenberth 1997). Historically, the definition of ENSO does not include precipitation

because of the limitations of stations (Ropelewski and Halpert, 1987), but recent work with satellites has confirmed that this phenomenon is a major driver of global precipitation variability (Haddad et al., 2004).

A major advantage of satellite-derived precipitation indices over more conventional ones is the description of the strength and position of the Walker circulation (Curtis and Adler, 2000). Under that assumption, Curtis and Adler (2000) derived

three satellite-based precipitation indices: the ENSO precipitation index (ESPI); El Niño index (EI); and La Niña index (LI). Precipitation anomalies are averaged over areas of the Equatorial Pacific and Maritime Continent -where the strongest precipitation anomalies associated with ENSO are found- to construct differences or basin-wide gradients (Curtis, 2008).

Figure 13 shows a comparison of GPCP, CONTROL, and POPEM for the ESPI, EI and LI indices.

Figure 13 about here

Unfortunately, CONTROL and POPEM cases have difficulty simulating the precipitation patterns associated with ENSO. Figure 13 shows that bias increases in 82-83 and 97-98 El Niño years. The same bias emerges when comparing the EI and LI

indices. In that case, the CESM model produces stronger El Niño/La Niña events than the observed data. Consequently, we can consider that CESM is unable to obtain a precise estimate of precipitation patterns, suggesting that current climate models are far from generating realistic simulations of the precipitation field (Dai, 2006).

Another widely used ENSO index is the Oceanic Niño Index (hereafter ONI). ONI was developed by the NOAA Climate

Prediction Center (CPC) as the principal means for monitoring, assessing and predicting ENSO (Kousky and Higgins, 2007). This index is defined as 3-month running-mean values of SST departures from the average in the Niño-3.4 region. It is computed from a set of homogeneous historical SST analyses (Kousky and Higgins, 2007, Smith et al. 2002).

Figure 14 about here

Figure 14 compares the ONI index for CPC, POPEM and CONTROL cases. It is clear from the figure, that POPEM
produces a more realistic representation of the ENSO, especially if we focus on the 1992-1999 period. POPEM also obtains
better results than CONTROL in the number of simulated el Niño events (see Table 1). The improvement is also noticeable
in the intensity. The CONTROL case exhibits an overly strong ENSO -a common bias in CESM (Tang et al., 2016)- but
POPEM reduces this bias (0.22$^{o}$ C versus 0.59$^{o}$ C).

Table 1 about here

Another important indicator is the mean duration of El Niño events. Table 1 shows that POPEM obtains better results
according to observations (11 months in CPC, 10 months in POPEM, and 19 months in CONTROL).

## 4. Conclusions and future work

Like all models, climate models are simplified versions of the real world and therefore do not include the full complexity of
the Earth system. Due to certain limitations, e.g. computational resources, or spatial and temporal resolution, climate models
have to make assumptions and resort to parameterizations.

One important simplification is to use prescribed forcings instead of dynamically modeling GHG emissions. However,
precise modeling of anthropogenic $CO_2$ emissions is important for climate change research as it allows sensitivity analyses to
be performed.

Here we present a new module of gridded $CO_2$ emissions that is coupled with CESM. The module, denominated POPEM,
computes anthropogenic $CO_2$ emissions by using population estimates as a proxy for disaggregating emissions beyond the
national level. POPEM makes CESM use dynamical emissions data instead of fixed concentration parameters.

In terms of population and emissions, the module compares well when validated with data. Thus, POPEM's estimates for the
1950-2000 period are in general agreement with population and emission inventories from the recent past. In spite of the
more realistic depiction of the actual emissions (Figure 3), issues persist. The performance of the model can be further
improved in places where population projections are difficult to model. For instance, POPEM tends to underestimate
emissions on the West Coast of the United States and the Anatolian Plateau, and overestimates emissions in China and
Japan.

When the POPEM module is coupled with CESM to generate climatologies, the ability to successfully model precipitation and surface temperature is preserved. Moreover, the results of 50-year simulations show that the dynamical modeling of emissions produced by POPEM results in slight but noticeable differences in the resultant precipitation regime and surface

temperature. Thus, dynamically modeling the emissions alters the ITCZ by reducing precipitation in the Southern Hemisphere and increasing it in the Northern Hemisphere. For particularly interesting areas, such as the El Niño-4 region, the POPEM outperforms the traditional approach.

Further work will be devoted to improving the modeling of those areas and hopefully minimizing some of the original biases

of the CESM model. These include the emergence of a double ITCZ in CESM simulations, which is a common bias for most climate models (Oueslati and Bellon, 2015), as well as SST simulated by climate models, which are generally too low in the Northern Hemisphere and too high in the Southern Hemisphere (Wang et al., 2014).

Current applications of the parameterization include evaluating the effects of changes on regional policies, and a better

understanding of the carbon cycle (Friedlingstein et al., 2006). Future work will be devoted to evaluating the climate response to alternative anthropogenic $CO_2$ emissions; to fully coupling Human-Earth subsystems; to increasing the spatial resolution of the simulations; and to refining the spatial and temporal distribution of emission estimates.

Although the version of POPEM presented here is already functional, this work is intended to be just the first step in fully

coupling socioeconomic dynamics with ESMs. This will include bidirectional feedback between Human and Earth systems and the simulation of societal processes based on the internal dynamics of the model instead of using external sources to make the projections. Only within a coupled global Human-Earth system framework can we produce more realistic representations of the Earth system capturing much of the counterintuitive feedback that is missing from current models (Motesharrei et al. 2016). The success of this approach will depend on the ability of scientists from different research fields

to work in an interdisciplinary framework of continuous collaboration.

**Author Contribution**

ANM and FJT contributed to experiment design, coding, analysis, manuscript writing and made the amendments suggested

by the referees. RMG contributed to manuscript writing and POPEM-CESM implementation in the UCLM supercomputing center.

**Competing interests**

The authors declare that they have no conflict of interest.

**Acknowledgements**

Funding from projects CGL2013-48367-P, CGL2016-80609-R (Ministerio de Economía y Competitividad, Ciencia e
Innovación) is gratefully acknowledged. ANM acknowledges support from grant FPU 13/02798 for carrying out his PhD.
We want to thank the five referees for their constructive comments and recommendations. Their comments have greatly
improved the manuscript.

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

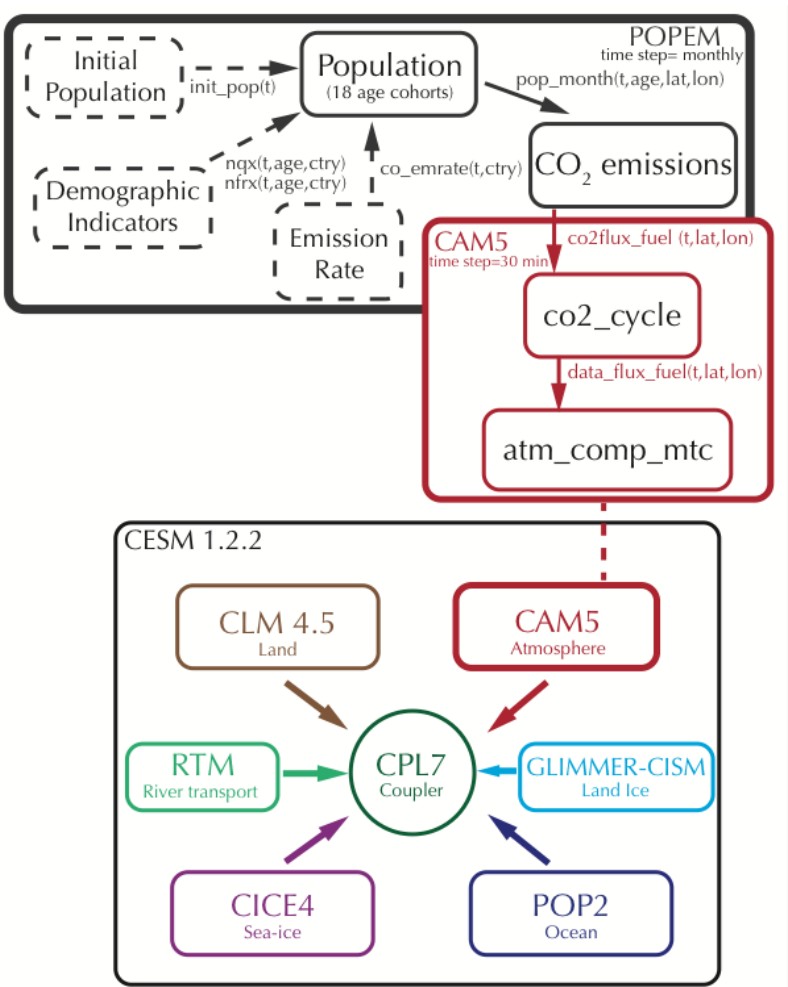

**Figure 1: Conceptual schema of the POPEM module coupled with the CAM5 atmosphere module. POPEM requires three input data sets to compute emissions (black dashed rectangles): initial population distribution; demographic parameters (age structure, death, and birth rates); and per capita emission rates by country. POPEM provides a 3D array (time, latitude, longitude) with emissions that are read by the *CO2_cycle* module and passed to the *atm_comp_mct* module which computes the total amount of $CO_2$ in the atmosphere.**

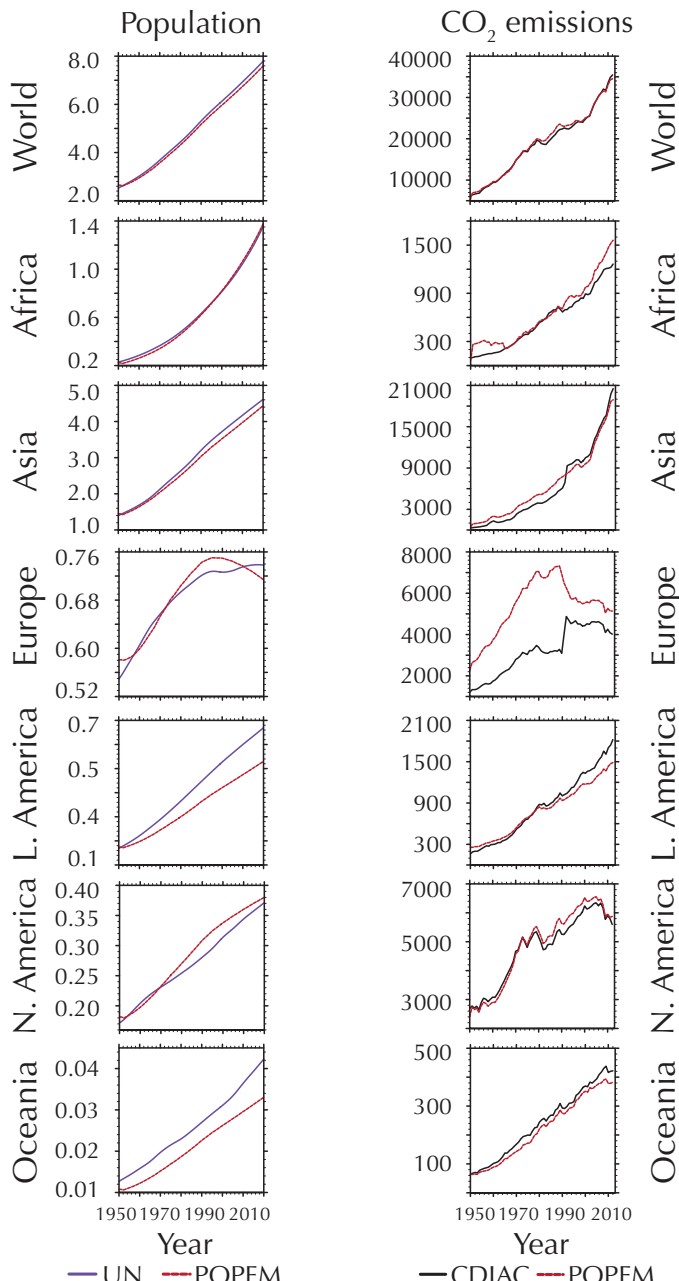

**Figure 2: Comparison of the population estimates for the years 1950-2020 (left column) and the historical CO₂ emissions estimates for the years 1950-2012 (right column). The first row compares global data, the second to seventh compare regional data (Africa, Europe, Latin America, North America and Oceania). In the left-hand column, the red line shows the estimates given using POPEM and blue indicates UN estimates. Values are given in thousand millions of people. On the right, the red line shows the estimates given using POPEM and the black indicates CDIAC estimates. Units are given in million metric tons.**

## Evolution of the CO$_2$ emissions in the POPEM model

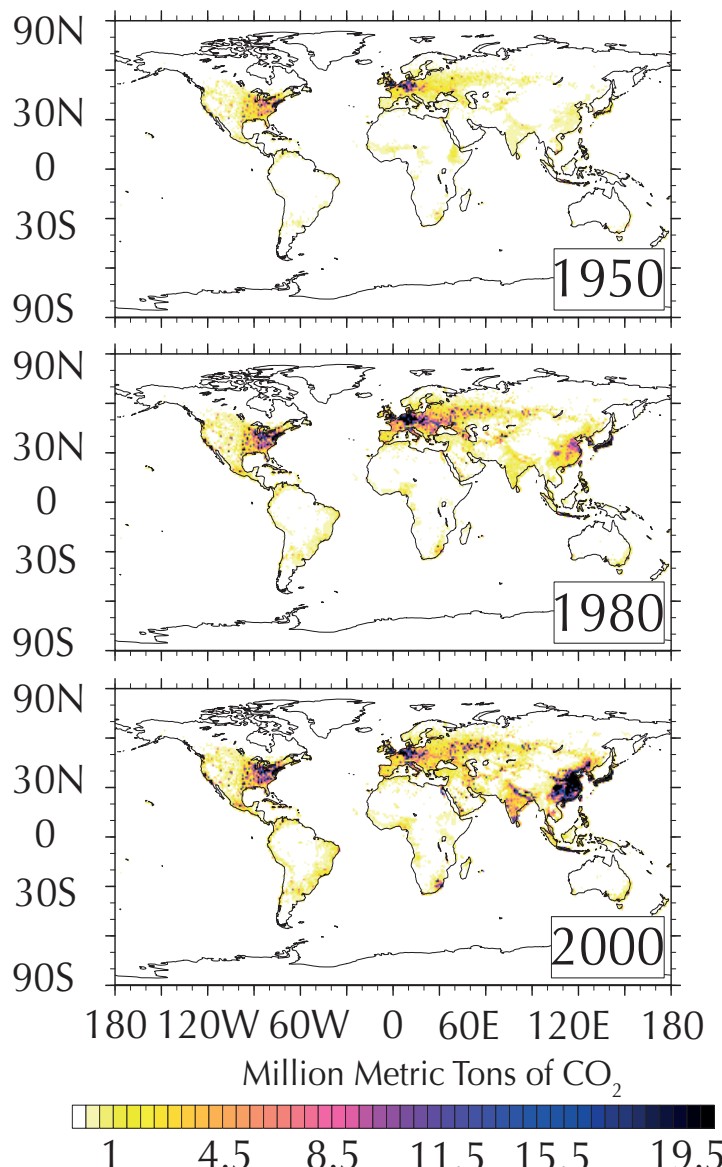

**Figure 3: POPEM CO$_2$ emissions estimates for 1950, 1980 and 2000. POPEM produces a gridded representation of anthropogenic CO$_2$ emissions using population dynamics and country per capita emissions derived from the CDIAC database. Values are given in millions of metric tons per year.**

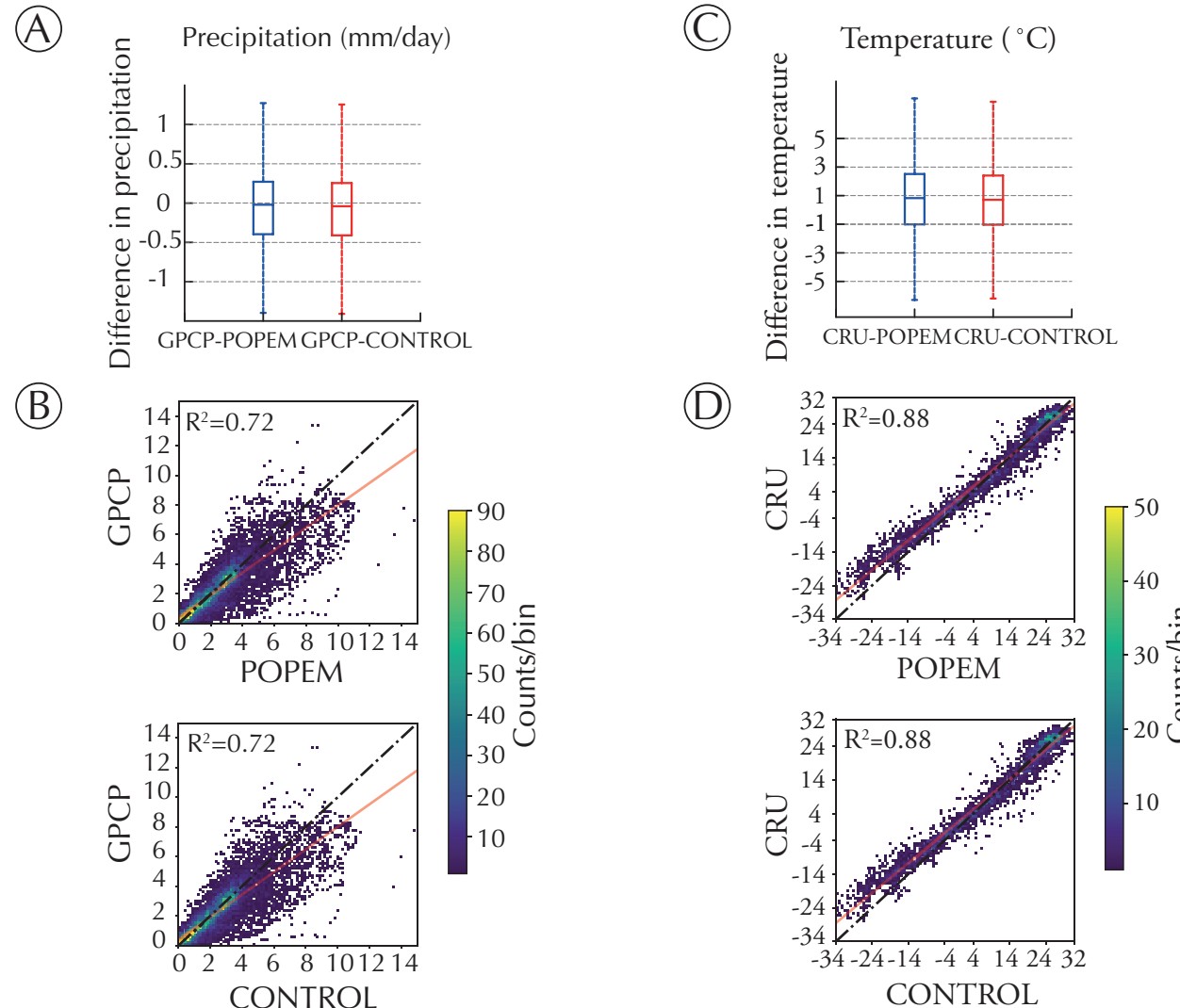

**Figure 4: Boxplots of CESM simulation bias for precipitation (A) and temperature (C). (B) Scatter plots comparing the annual mean precipitation (1980-2000) at every grid point for GPCP and CESM simulations (POPEM and CONTROL). (D) Scatter plots comparing the annual mean temperature at every grid point for CRU and CESM simulations (POPEM and CONTROL). Units are in mm/day (precipitation) and in degrees Celsius (temperature).**

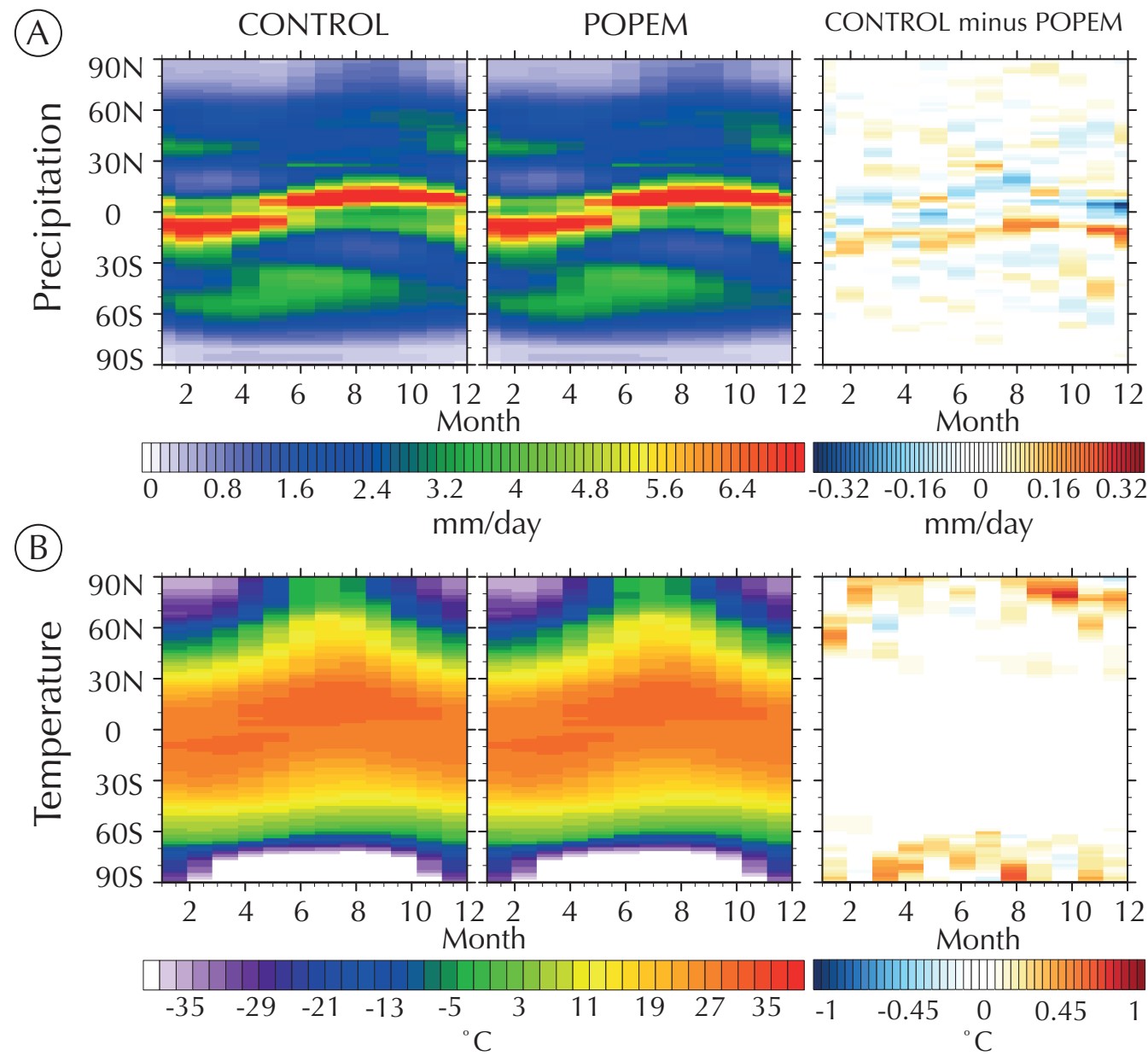

**Figure 5: Latitude vs time plots for precipitation (A) and surface temperature (B). For absolute difference graphs, blue represents higher values in POPEM and red represents higher values in the CONTROL. Units are in mm/day for precipitation and in Celsius for temperature.**

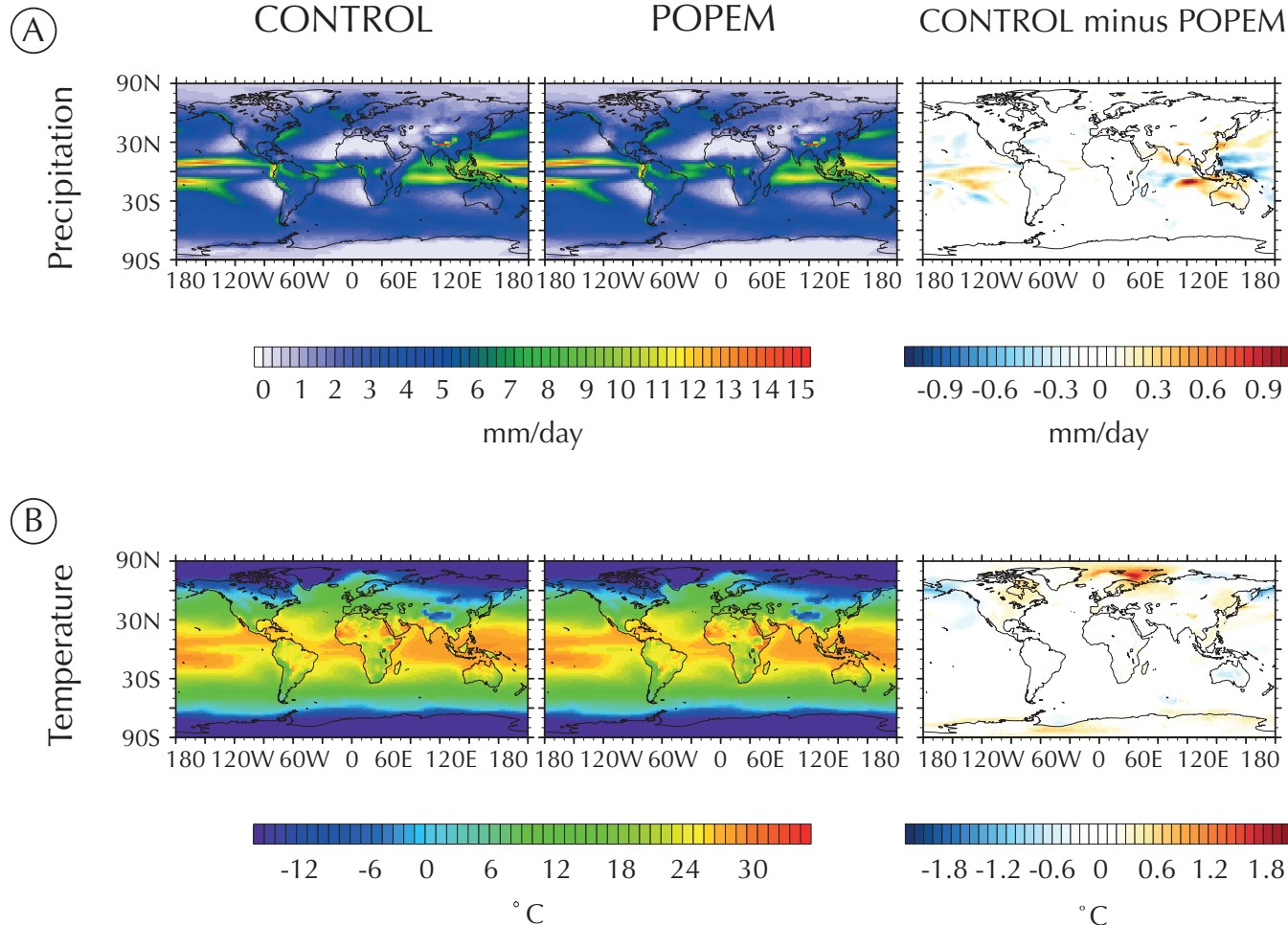

**Figure 6: A comparison of global annual mean precipitation (1950-2000) for the CONTROL and POPEM (A). (B) is a comparison of annual mean surface temperatures. The maps in the right-hand column show the absolute differences between the simulations (CONTROL minus POPEM). In these, blues represent higher values in POPEM and reds represent higher values in the CONTROL.**

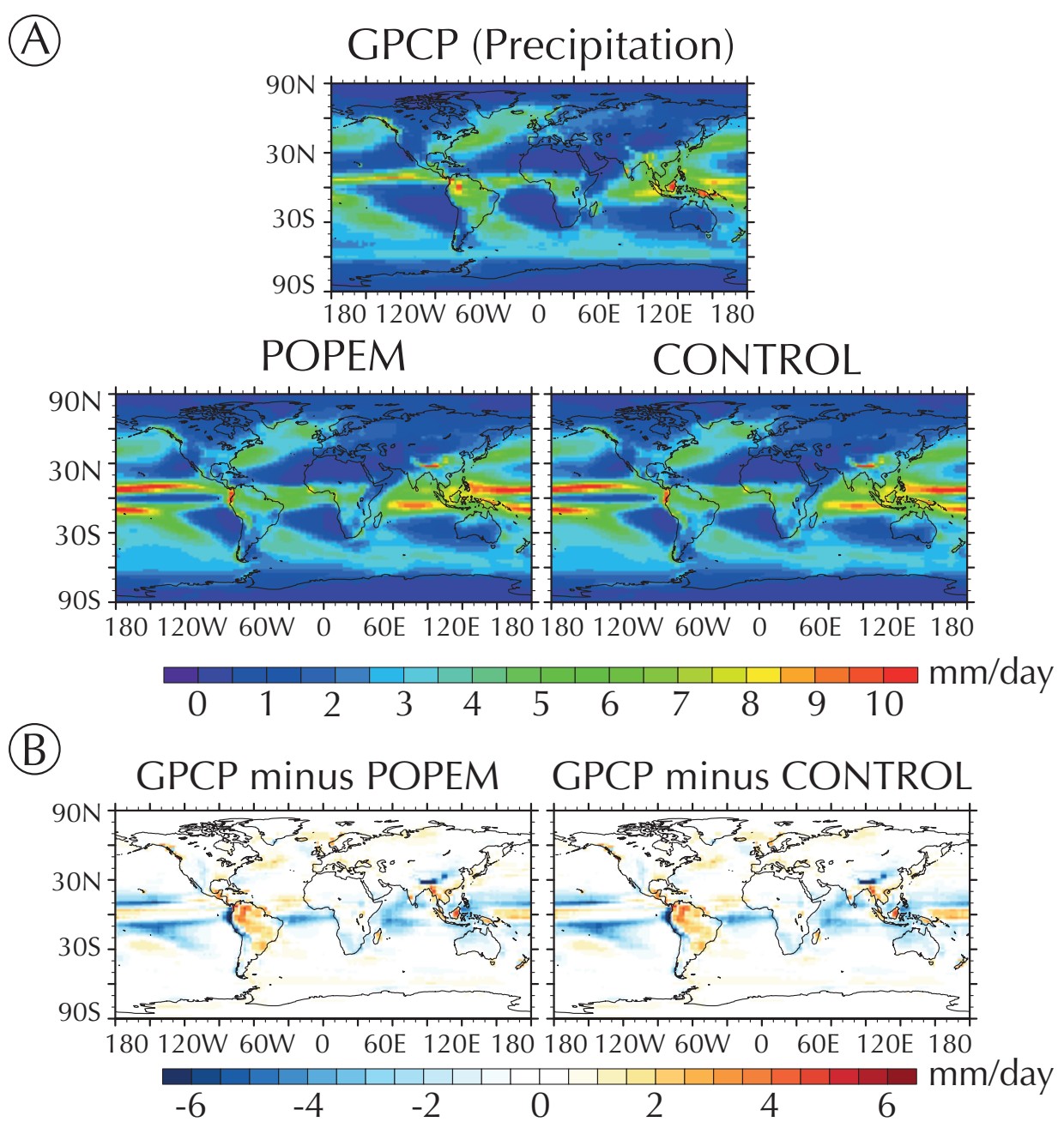

**Figure 7: A comparison of the global annual mean precipitation (1980-2000) as simulated by the CESM (POPEM and CONTROL) model and GPCP observational database. (A) Global annual mean precipitation maps for GPCP, POPEM and Control. (B) Absolute difference maps. Units are in mm/day.**

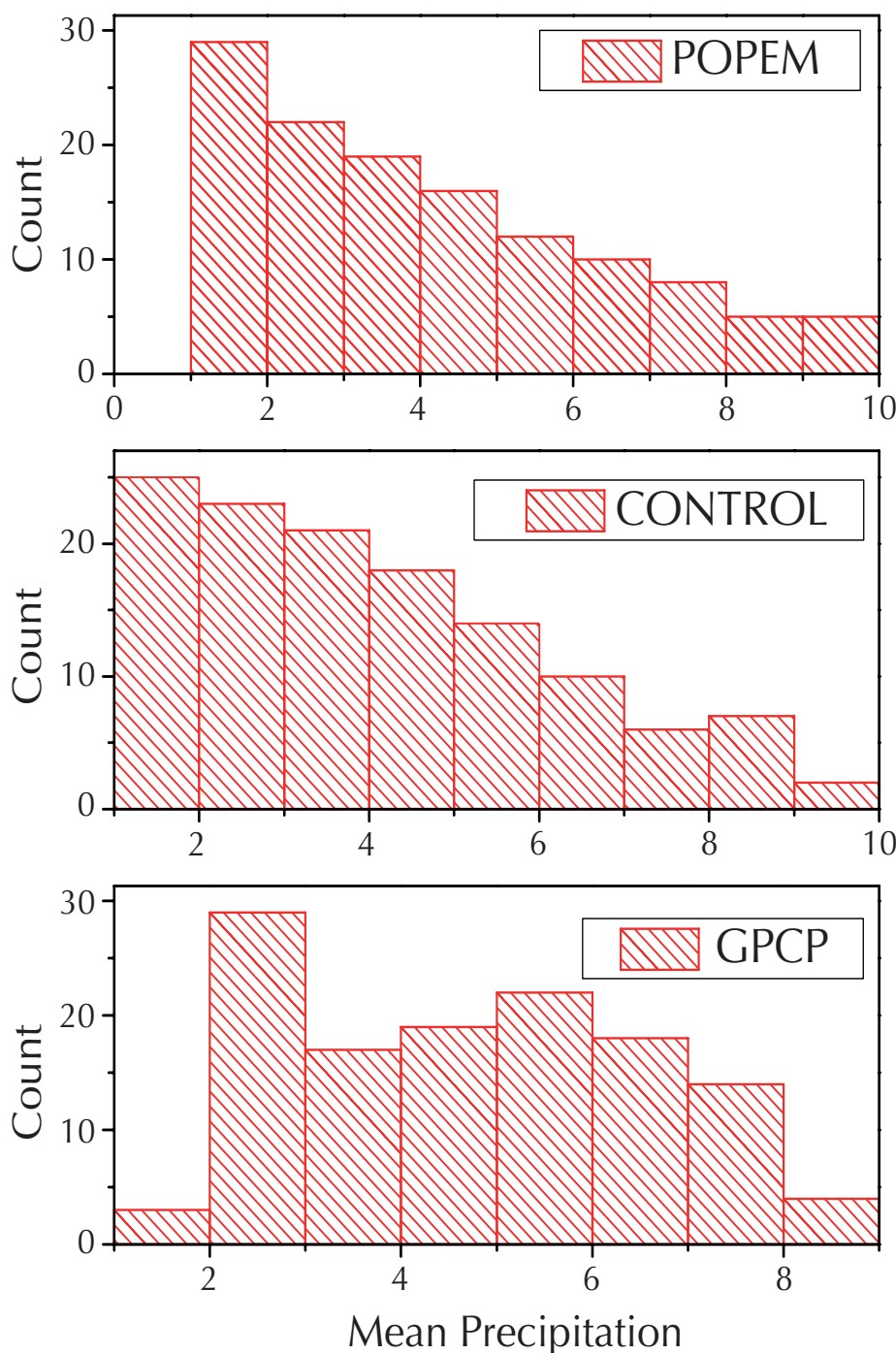

**Figure 8: Histograms of the mean precipitation in the El Niño-4 area (5N-5S, 160E-150W) using the POPEM parameterization (top), the standard forcing approach (CONTROL, middle), and the reference GPCP estimates (bottom).**

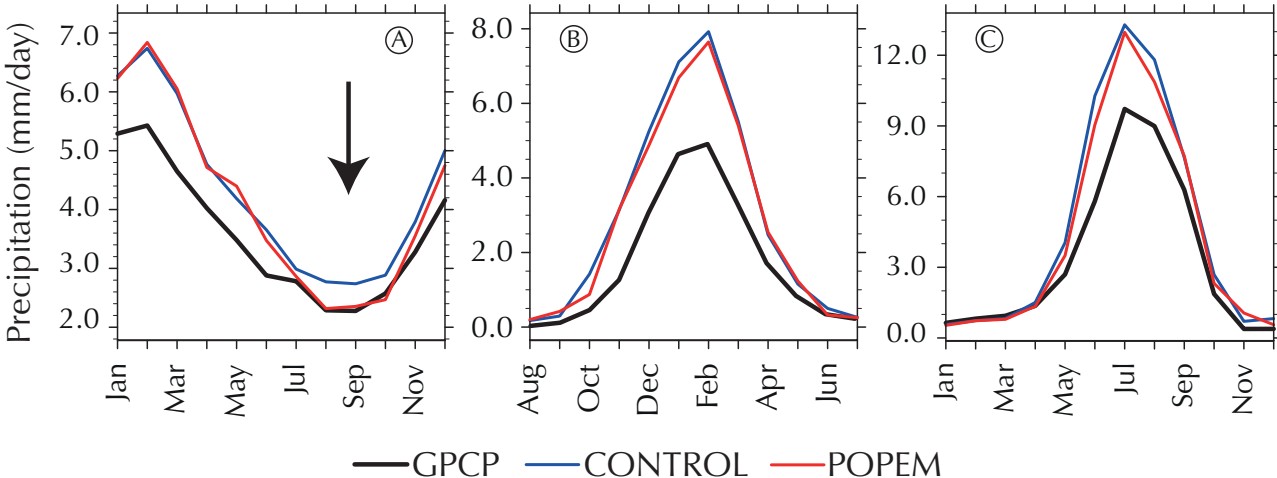

**Figure 9: Monthly precipitation (1980-1999) based on GPCP, CONTROL and POPEM for three of the regions with important biases in CESM. (A) shows precipitation for the area affected by the double-ITCZ bias in the Southern Hemisphere (20S-0, 80E-100W); (B) for Australia Top End (30S-10S, 128E-140E); and (C) for the Tibetan Plateau (22N-32N, 78W-92W). The black line represents observations (GPCP), the blue line is the CONTROL case, and the red line is the POPEM case. Units are in mm/day. The arrow indicates the improvement of the POPEM model.**

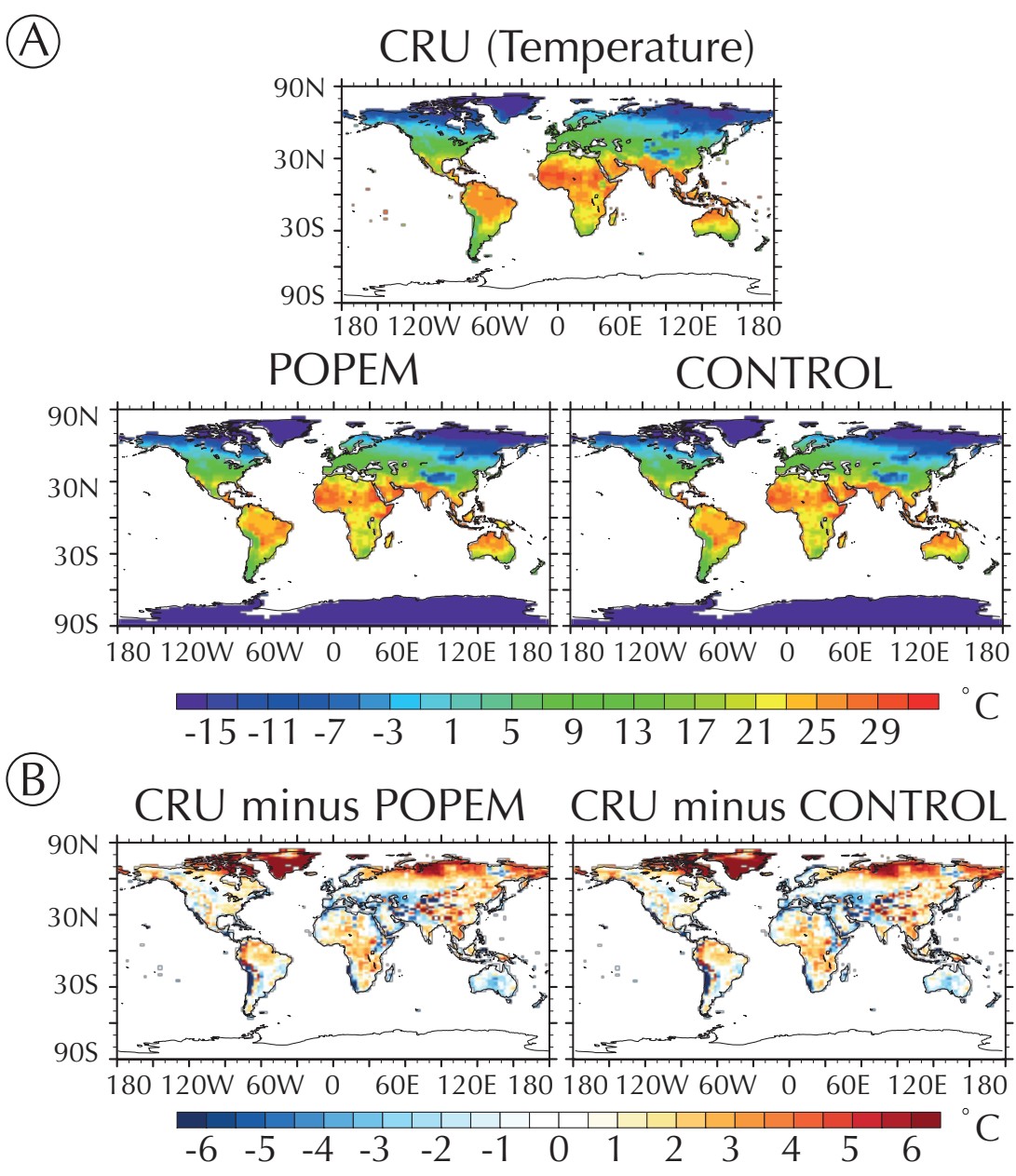

**Figure 10: A comparison of the annual mean temperature (1950-2000) as simulated by the CESM model (POPEM and CONTROL) and CRU observational database. (A) Global annual mean temperature maps for CRU, POPEM and CONTROL. (B) Absolute difference maps. Units are in degrees Celsius.**

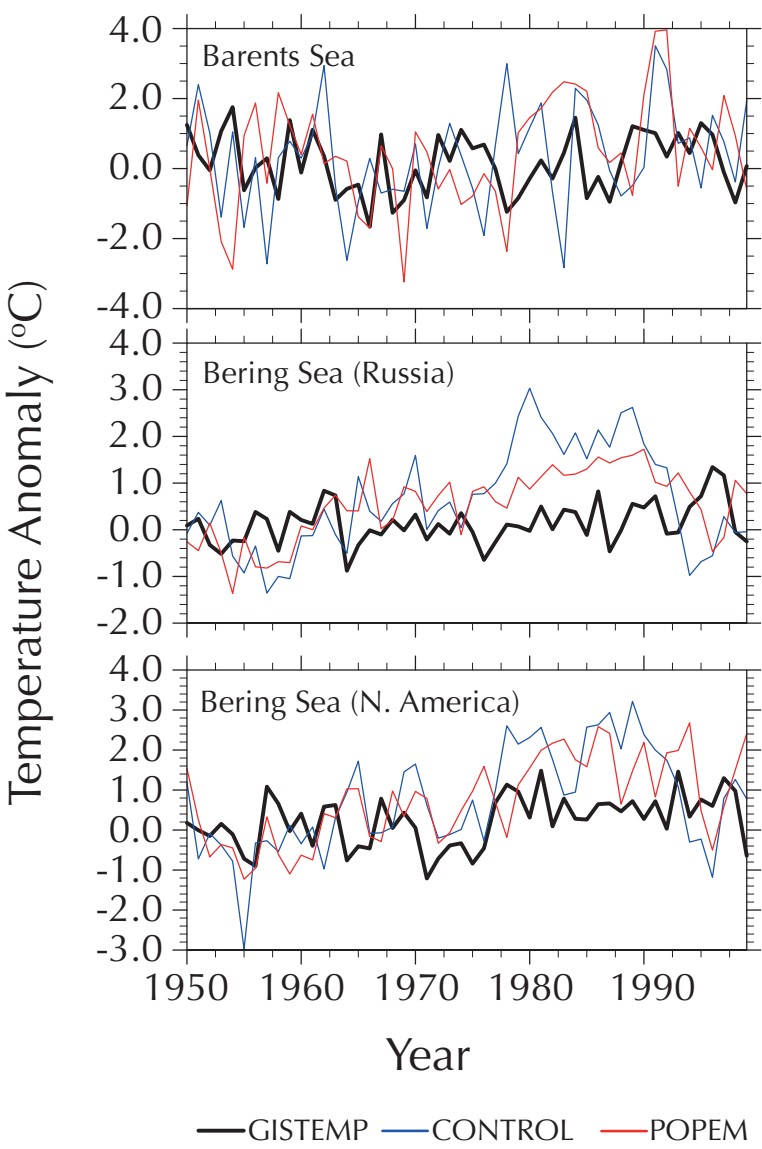

**Figure 11: A comparison of the annual mean surface temperature anomaly between GISTEMP, CONTROL and POPEM from 1950 to 1999. (Top) represents the Barents Sea (68N-80N, 19E-68E); (middle) Russian part of the Bering Sea (50N-65N, 150E-180E); and (bottom) American part of the Bering Sea (50N-75N, 140W-180W). The black line represents observational data (GISTEMP), the blue line is the CONTROL case, and the red is the POPEM case. Anomaly was referenced to 1951-1980 period.**

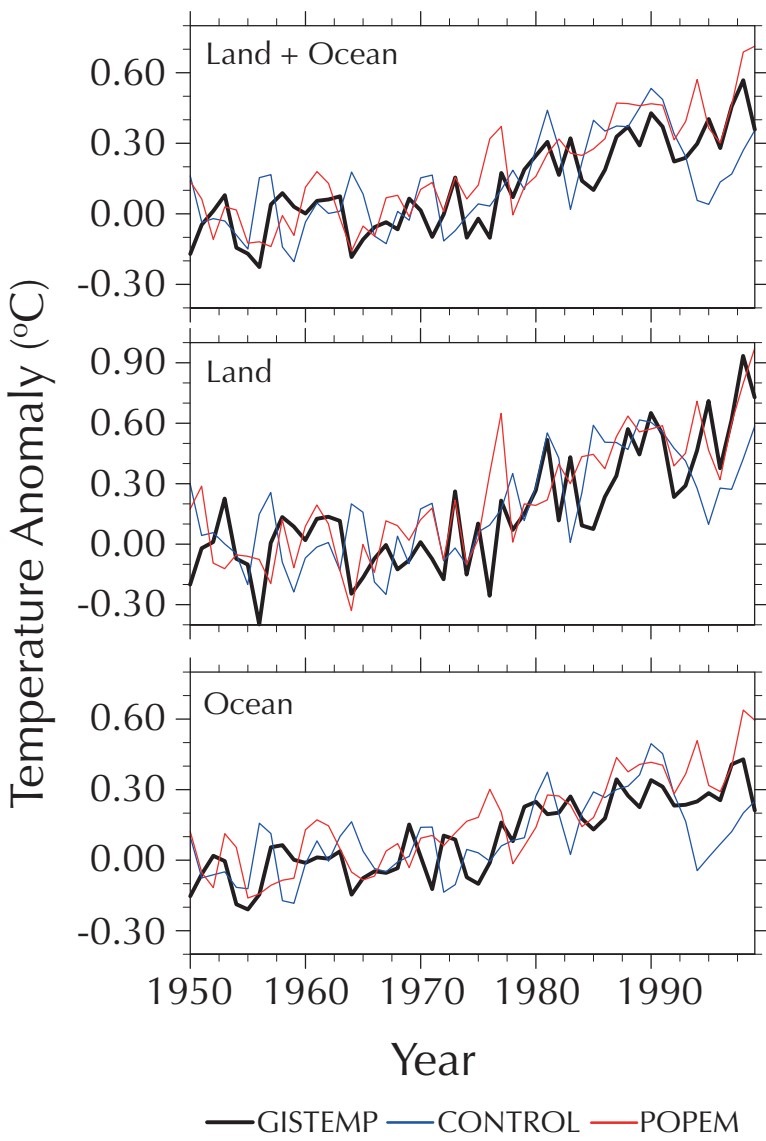

**Figure 12: A comparison of the global annual mean surface temperature anomaly between GISTEMP, CONTROL, and POPEM from 1950 to 1999. (Top) global; (middle) land; and (bottom) ocean. The black line represents observational data (GISTEMP), the blue line is the CONTROL case, and the red is the POPEM case. Anomaly was referenced to 1951-1980 period.**

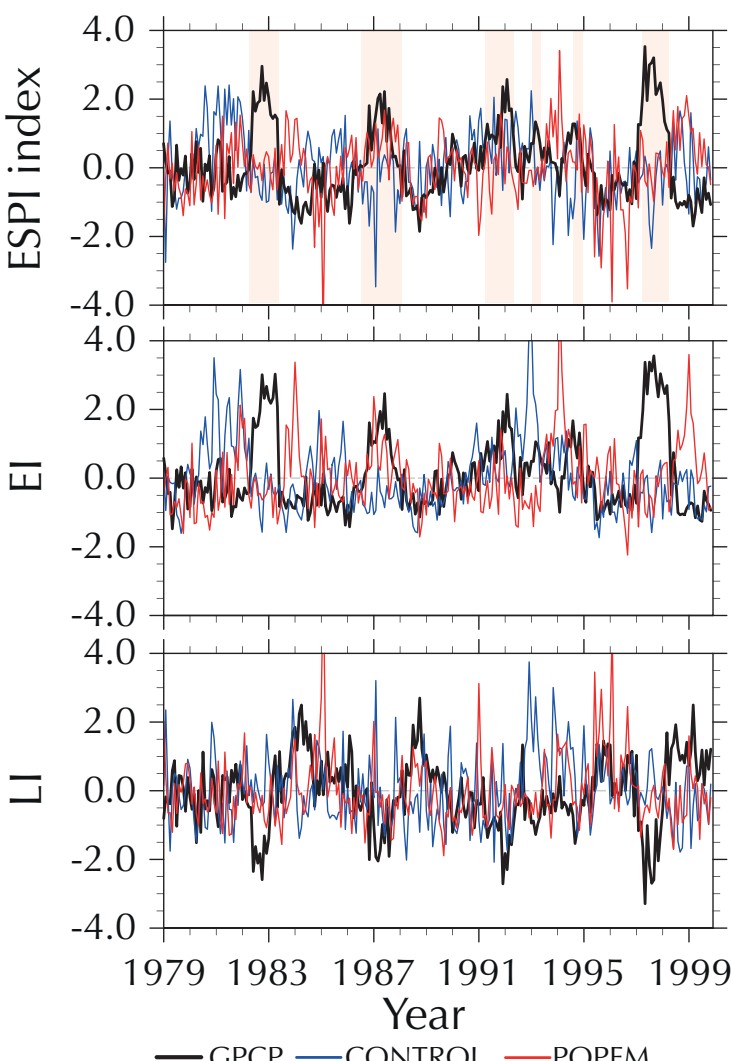

**Figure 13: Time-series of precipitation anomalies for the ENSO region after Curtis and Adler (2000). (Top) ENSO Precipitation Index (ESPI); (Middle) El Niño Index (EI); and (Bottom) La Niña Index (LI). The Black line shows GPCP data, the blue line is the CONTROL case, and the red line is the POPEM case. Orange shading denotes El Niño years defined as consecutive months (minimum 3) with NIÑO3.4 sea surface temperature anomalies (5N–5S, 170–120W) greater than +0.5°C.**

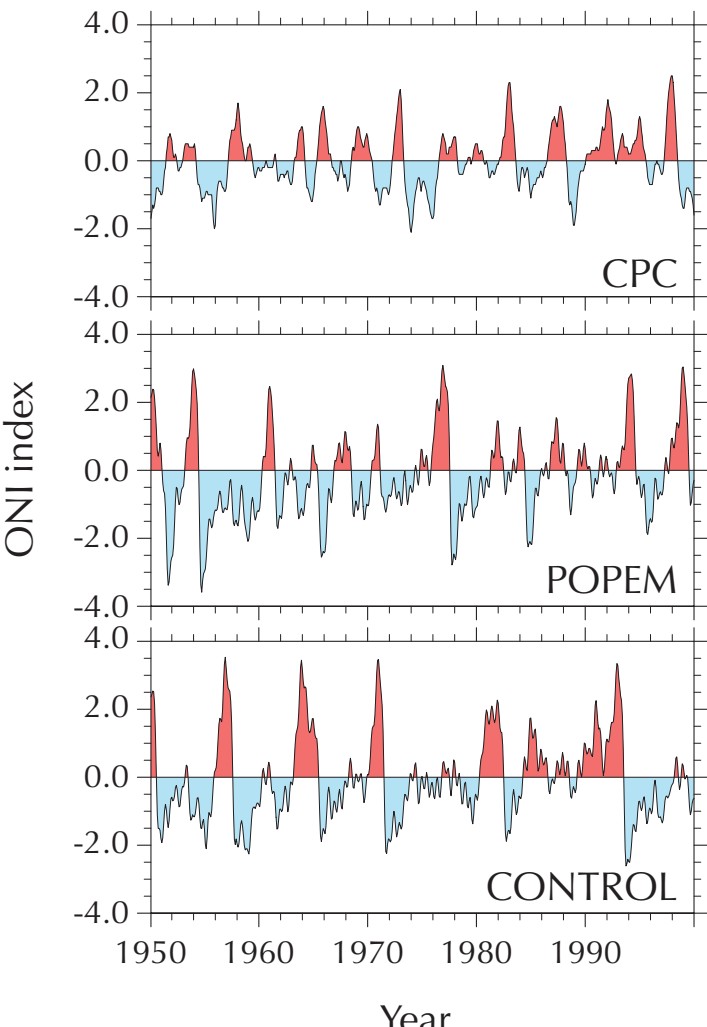

**Figure 14:** Comparison of the Oceanic el Niño Index (ONI) for CPC (top), POPEM (middle), and CONTROL (bottom) cases. El Niño and La Niña are defined according to Kousky and Higgins (2007): 3-month running mean with anomalies greater than $+0.5^{o}C$ (or $-0.5^{o}C$) for at least five consecutive months in NIÑO3.4 region. The base period for computing SST departures is 1971–1999.

**Table 1**. Comparison of the ONI index for the period 1950-1999. The table compares the ability of the models to reproduce the number, strength, and duration of el Niño events.

| Source | Number of events | Agreement [1] | Disagreement [2] | Intensity Bias $_{avg}$ [3] | Duration $_{avg}$ [4] |
|---|---|---|---|---|---|
| CPC | 14 | | | | 10.3 |
| CONTROL | 7 | 33 | 121 | $0.59\,^{\circ}$ C | 19.4 |
| POPEM | 10 | 37 | 121 | $0.22^{\circ}$ C | 11.4 |

[1] The number of months that CPC and CESM agree on El Niño. [2] Disagreement defined as the number of months where CPC and CESM obtain opposite results. [3] Intensity: (|CESM ONI| – |CPC ONI|)/number of cases (units in degrees Celsius). [4] Mean duration of El Niño event (in months).

