# Peer review of "Improving the representation of anthropogenic CO2 emissions in climate models: impact of a new parameterization for the Community Earth System Model (CESM)"

_Earth System Dynamics, 2018_

## Referee Comment (RC1) · Anonymous Referee #1 · 22 Mar 2018

The paper is very interesting, novel and merits immediate publication. The approach is that of a 'proof of concept', but the idea behind the research is extremely interesting and worth of attention by the community.

However, I believe the authors must touch upon several topics in order to improve the paper. Specifically:

-Last part of section 3.1 needs further explanation. Please expand the section and provide more information about potential applications. I think that is an important part

of the paper (probably the most important part), and it is a pity that the authors give just such a swift account of the topic. Âă

-Given the large number of papers using CESM I think more attention should be devoted to previous work using this model. Please add several references to show how CESM has been used, including merits, shortcomings and the like.Âă

-The double ITCZ issue needs referencing. Who did first mention that? Without such reference it seems that that feature is a novel observation from the authors, which I think it is not.

-The following sentence is confusing to me. ÂăThe improvements of POPEM for the El Niño-4 area show that detailed, dynamical modeling of GHG emissions is important for more precisely quantifying precipitation in dry areas, which validates the main hypothesis of the paper.ÂăPlease explain what do you mean by that.

- Please, check the references to the figures in the text (Figs. 4 and 5)

- Please, check the place where you put the definition of some acronyms (e. g. ITCZ -you use it on page 7 and is defined in page 10-, SST -similar-).

---

## Referee Comment (RC2) · Anonymous Referee #2 · 17 Apr 2018

General Comments:

This paper highlights the importance of grid point scale modeling of anthropogenic pollutants, especially CO2, and the integration of such modeling through a new module called "Population Parameterization for Earth Models" (POPEM). The module is integrated into a highly distributed climate model like Community Earth System Model (CESM). The authors present clearly and adequately the added value of their contribution (POPEM) to the model simulations and underline its impact to the climate predictions of both precipitation and temperature.

Minor comments:

Figures 6B and 8B do not illustrate clearly any differences between GPCP – CONTROL, GPCP – POPEM and CRU – CONTROL, CRU – POPEM respectively. Maybe the authors should consider an alternative way to show the differences.

---

## Referee Comment (RC3) · S. Hristova-Veleva (Referee) · 18 Apr 2018

**Review of original submission**

**Paper Title:** Improving the representation of anthropogenic $CO_2$ emissions in climate models: a new parameterization for the Community Earth System Model (CESM)

**Authors**: Andrés Navarro1, Raúl Moreno1 5 and Francisco J. Tapiador1
1Institute of Environmental Sciences (ICAM), University of Castilla-La Mancha, Toledo, 45004, Spain.
Correspondence to: A. Navarro (Andres.Navarro@uclm.es)

**Overview and recommendations:**

The paper addresses important questions regarding improving the performance of Earth System Models (ESM) – the important tools to study and understand the complexity of the Earth's climate. Improving these models is a major goal of the science community as they can be a very valuable tool in studying the response of the Earth's system to anthropogenic forcing, providing guidance to policy makers.

In particular, the paper investigates the impact of a new parameterization of CO2 emissions that the authors have recently developed, called the POPEM module (POpulation Parameterization for Earth Models). POPEM presents an important advancement in the way CO2 emissions are modeled, as it accounts dynamically for the changing emissions. Like previous research, POPEM uses population data as proxies for emission. What is unique to this new parameterization, though, is that it models the evolution of the population while previous research has relied on historical data, hence not being dynamical, preventing them from making reliable predictions for the future emissions and the response of the climate system.

Using this new parameterization (POPEM) presents an important advancement and this makes the described research very valuable. However, before going forward one have to evaluate the performance and assess the impact of the new parameterization. Indeed, this is the goal of this paper.

The paper begins by describing what is unique about POPEM.

It then validates the stand-alone performance of POPEM by comparing its predication over a past 63 (and 70) -year period to existing data. The comparison is done globally but also by several regions. This validation is done in two ways: by comparing forecasted to observed population growth rates; and by comparing the forecasted to observed emission rates. The results show that despite the difficulty of predicting non-linear trends in the growth of population and emissions, POPEM preforms quite well. These comparisons give credibility to the POPEM forecasts, hence to its use in forecasting future scenarios.

Next, the paper uses a coupled ESM, the Community ESM (CESM) to evaluate the impact of POPEM.  The evaluation focusses on the impact of POPEM on two very important, and difficult to predict, parameters of the Earth's system - the precipitation and the sea surface temperature (SST).  The evaluation is done in two ways:

- by comparing the results from a control run (using global $CO_2$ concentration parameters that I believe are homogeneous – this needs clarification) to those from POPEM.  This choice of model setups highlights the value of POPEM as it predicts the population (and the emissions) in every grid point, showing the impact and the importance of the spatial variability.
- By comparing both control and POPEM forecasts to actual observations (over a 20-year period for precipitation and 50-year period for SST).

The paper finds that:

- The global predictions for both parameters compare to the observations in a very similar way for the CONTROL and the POPEM simulations.  Hence, the more realistic POPEM parameterization "does no harm".  This is an important test and conclusion because it is occasionally the case that including more realistic parameterizations might degrade the performance of the forecasts for certain parameters.  This is because often the models are "tuned" to predicting some of the parameters, giving the right answer for the wrong reason, and impacting negatively the forecasting of the non-tuned parameters when the more realistic parameterizations are employed.
- More importantly, the paper finds that using POPEM results in regional differences between its forecasts and that of the control run.  Comparison to observations seems to suggest the POPEM produces better regional distribution of the precipitation.  This is a very important conclusion, in my view.  It does not seem to be well highlighted in the paper summary.

Overall, the paper addresses a very important topic.  The approach is sound and uses a very good modeling framework.  There is a very extensive set of references.  The paper is presented in a fluent and precise language.

However, there are several places where the paper could be improved, as detailed below.

**Because of all that, I propose the paper be accepted with minor revisions.**

**Detailed comments and suggestions for modifications**

**Title**: The current title is: "*Improving the representation of anthropogenic $CO_2$ emissions in climate models: a new parameterization for the Community Earth System Model (CESM)*"

I would suggest a modification to read "*Improving the representation of anthropogenic CO2 emissions in climate models:* **Impact of** *a new parameterization for the Community Earth System Model (CESM)*"

The reason is that main goal of the paper is not to describe the new parameterization but to evaluate its performance and impact.

**Abstract**
- "*The results show that it is indeed advantageous to model CO2 emissions and pollutants directly at model grid points rather than using the forcing approach*". Please, reword as it is not clear (at this point) what is this forcing approach.

**Introduction:**
- The reader would benefit from a more detailed description of the existing approaches to modeling CO2 emissions. What I gather from the paper is the following: there are two basic approaches that models use to account for CO2 forcing:
    a) using globally homogenous forcing;
    b) using non-homogenous, grid-point specific forcing. This one can be applied in several ways:
        1. using Representative Concentration Pathways (RCPs) that "*are not fully-integrated socioeconomic parameterizations, but rather estimates for describing plausible trajectories of human climate change drivers .... They provide simplified accounts of human activities and processes, including population density and economic development, from non-coupled Integrated Assessment Models (IAMs;)*" Question: are these parameters location-specific? This is what I am understanding.
        2. the proposed here POPEM model being integrated into a fully coupled model. This is similar to RCPs but: uses a coupled model; uses a dynamic model for the prediction of population and emissions.
    c) Is my understanding correct???
    d) If so, I would suggest two possible modifications:
        1. Use some wording or structure as what I've described above
        2. Space-permitting, create either a small table or a flow diagram that shows these different levels of sophistication
- P. 2, lines 25-30 – It says: "*Given the highly non-linear character of the processes involved, it is not unreasonable to assume that location is significant, and the spatial and time distribution of these emissions may affect global climate*" – a bit unclear. Might be better to say "*, it is not unreasonable to assume that* specifying (or accounting for) geographical variability *is significant*"
- P. 3, lines 2-4: "*The aim of this paper is to show that this grid point scale modeling of anthropogenic CO2 emissions (and other pollutants) represents an improvement, and that two important variables, namely global precipitation distribution and surface temperature, are not negatively affected by this more-detailed approach.*" While this is true I believe this is a rather weak statement regarding the benefits of using POPEM-type parameterization of emissions forecasting. I believe the authors are in a position to make a stronger statement,

namely: including the POPEM dynamical forecasting approach that accounts for the spatial and temporal variability of the emission sources, leads to better representation of the geographical variability of the precipitation.

- Space-permitting, I would suggest that the **Introduction** ends with a short description of the outline for the following presentation. Something like: " *the following sections outline: the unique features of POPEM; the validation of the POPEM stand-alone performance; the framework for evaluating the impact of POPEM – incorporation into CESM and framework for testing; the comparison between a control run and a POPEM-specific one: evaluating the differences between the two; evaluating how each compares to observations; discussions; summary and conclusions;*" This would give the reader a clear structure of the paper to follow and will make it easier to highlight the contributions of the paper.

**Section 2.2**

- currently there are sections 2.2 and 2.2.1 but not 2.2.2 or more. It seems that there is no need for 2.2.1. If there is no 2.2.2. I would suggest the following: "2*.2 POPEM specifics and validation*", followed by "*2.2.1 POPEM parameterization model overview: Unique features*" and "*2.2.2 POPEM trend verification*". Of course, this is just a suggestion.
- P. 6, lines 8-9 – " *Our control case used global CO2 concentration parameters (standard procedure in ESMs), while the POPEM case used geographically-distributed CO2 emissions data*" - is the control using homogeneous $CO_2$ concentrations ? I am pretty sure this is the case but it might be better to say it this way.

**Section 3.1**

- P.7, line 23 – it appears that figures 6C, 6D, 8C and 8D are referenced before figures 4 and 5 (and the figure 8 is referenced before Fig.7). This should not be the case. The figures should be referenced in order. However, it seems that this is because the current order of the discussions here might need to be modified. Below is what I mean.
    a) Maybe the order should be:
        1. Test for "no harm" – figures 6C-6D and 8C-8D show that.
        2. Compare the CONTROL to the POPEM simulations to see where exactly they differ.
        3. Compare both the CONTROL and the POPEM CESM simulations to the observations, looking at regional distributions. The comparison in steps 2 and 3 brings up the impact of the POPEM geographically-aware CO2 emissions on the geographical distribution of the precipitation, highlighting the positive impact POPEM has (especially in step3).
    b) Steps 2 and 3 could be switched – depending on what the authors think.
    c) I want to point out that the proposed change in the order of the presentation is just a suggestion for the authors to consider.
- P.8, lines 2-3: "*It is clear from the figure that POPEM does alter the spatial pattern of precipitation and exerts a definite effect on the climate pattern, as the module reduces the otherwise exaggerated ITCZ precipitation in the Southern Hemisphere (South East Asia and Australia)*." Do you have a reference that it was exaggerated?? If so, then this is a very strong point that needs to be emphasized. Also, do you mean Fig. 4 or Fig. 5? Please, specify.

- P. 8, lines 7-8: "*There are also important differences in precipitation in the 30N-30S band. Here POPEM reduces model bias, especially in the Southern Hemisphere and on the Tibetan Plateau.*"   How do we know that the model bias is reduced?
- P. 8, line 9-10: "*On the other hand, POPEM departs from the control simulation in the*
- *10 Asia-Pacific region between 10N-10S.*" Is that good or bad?  How do we know?
- *P*. 8, line 31 – "(*Q1 and Q3 remain between ± 0.4 mm/day*)."  Please, define Q1 and Q3.

---

## Referee Comment (RC4) · Anonymous Referee #4 · 22 Apr 2018

The authors developed a novel POPEM parameterization and applied it to CESM to enhance the realism of global climate modeling by improving the direct representation of human activities and climate. They argued that modeling $CO_2$ emissions and pollutants directly at model grid points is a better approach. As such, their new approach will help understand the potential effects of localized pollutant emissions on long-term global climate statistics, thus assisting adaptation and mitigation policies.

The topic is interesting and the approach is provoking. However, I am not quite convinced by the validation part (Part 3.2). I therefore recommend major revision.

First, I cannot find a remarkable improvement using POPEM based on the comparison of precipitation and temperature biases. There are some differences between POPEM and CONTROL but these differences are buried in the large biases in either set. It is true that observations have uncertainties and a new parameterization does not have to improve the model performance in every aspect. Nevertheless, could the authors show some improvements more robust than the current ones (precipitation and temperature) for validation? Maybe TOA radiation balance, ENSO index, Arctic sea ice, etc?

Actually, I am somewhat interested in the Arctic sea change. It is known that climate models (like CESM CONTRL) cannot capture a rapid observed decline of Arctic sea ice during recent decades. In Fig. 5(B), POPEM is colder than CONTROL over the Barents Sea area. Will this mean that Arctic sea ice decline in POPEM is even slower than that in CONTROL?

Besides, to be consistent with GPCP, the authors may want to use a globally (land+ocean) covered temperature dataset GISTEMP (https://data.giss.nasa.gov/gistemp/) to examine temperature bias.

---

## Referee Comment (RC5) · S. Motesharrei (Referee) · 5 May 2018

2018 APR 15

Review of the manuscript "Improving the representation of anthropogenic CO2 emissions in climate models: a new parameterization for the Community Earth System Model (CESM)" by Andrés Navarro, Raúl Moreno, and Francisco J. Tapiador, submitted to the Journal Earth System Dynamics, European Geosciences Union (EGU).

———————————————

[Figure]

Decision:

Because of the importance of the topic, I would recommend the publication of this manuscript after major revisions in the presentation of the work as described below.

————————————————

General Comments:

The authors acknowledge (but not completely clearly) a major shortcoming of the Earth System Models (ESMs) and Integrated Assessment Models (IAMs). Even though the Human System has become the dominant driver of most components of the Earth System since about 1750, and especially since about 1950, IAMs use independent, exogenous projections of the Human System (HS) variables in order to drive ESMs to create future projections. Not including essential bidirectional feedbacks between ES and HS can lead to missing important dynamics that is critical to the sustainably of our planet and people. This problem is discussed in detail in the "Modeling Sustainability" paper by Motesharrei et al. [2016]:

Motesharrei, Safa, Jorge Rivas, Eugenia Kalnay, Ghassem R. Asrar, Antonio J. Busalacchi, Robert F. Cahalan, Mark A. Cane, et al. "Modeling Sustainability: Population, Inequality, Consumption, and Bidirectional Coupling of the Earth and Human Systems." National Science Review 3, no. 4 (December 11, 2016): 470–494. https://doi.org/10.1093/nsr/nww081.

The manuscript is closely related to a recently published paper by the same team of authors (and, unfortunately, there is much overlap with that already published work):

Navarro, Andrés, Raúl Moreno, Alfonso Jiménez-Alcázar, and Francisco J. Tapiador. "Coupling Population Dynamics with Earth System Models: The POPEM Model." Environmental Science and Pollution Research, September 16, 2017, 1–12. https://doi.org/10.1007/s11356-017-0127-7.

These two papers take a step toward including at least parts of the Human System

(human population and emissions) explicitly in the ESMs, however, the somewhat inaccurate presentation of the work (and occasional over-statements) may lead to readers' confusion about the extent and novelty of this work. During my initial reading of the manuscript, I was very impressed by the model and thought that it is a bidirectionally coupled Human System + Earth System Model. (It seems Anonymous Referee 3 has this same impression.) But upon further reading of the manuscript as well as Navarro et al. [2017], I realized that POPEM is essentially a demographic projection model (although it uses dynamic variables for age cohorts) that is used to drive CESM. By contrast, I believe the use of local population projections to project emissions at each grid point is novel, and is advantageous to the current practice of using global emissions projections to drive ESMs.

———————————————————

Suggested Revisions:

The other three referees already provide many helpful, important suggestions to improve the manuscript. Here, I outline some additional suggestions to help accurately present the model, its value for the Earth System modeling community, and possible future steps that needs to be taken by the modeling community to make the projections of the "Earth–Human System Models" more realistic. I do not ask for any changes to model, since such changes would require major effort and could be implemented in future versions.

(A) Clarify that POPEM is, after all, a demographic projection model. It is true that its 18 age cohorts are dynamic variables, however, they still change based on exogenous fertility and mortality rates. (POPEM does not model Migration, which has become a major driver of population change, especially recently.) These rates are projected into the future using statistical methods such as in the UN Population Projections. Therefore, the projections using POPEM could not be much different from traditional demographic projections, as can be seen from comparisons of POPEM to UN projections in Navarro

et al. [2017]. I believe indeed POPEM cannot properly capture demographic change details for some regions and for certain age cohorts. Therefore, the value-added from this 'dynamic' population model is limited, at least from a demographic perspective.

(B) Because ES and other components of the HS do not feedback onto the demographic variables in POPEM, POPEM will not be able to capture non-trivial dynamics that can arise due to such bidirectional feedbacks [Motesharrei et al., 2016]. For basic examples of how these bidirectional feedbacks (in a minimal model) can lead to surprising behavior, see:

Motesharrei, Safa, Jorge Rivas, and Eugenia Kalnay. "Human and Nature Dynamics (HANDY): Modeling Inequality and Use of Resources in the Collapse or Sustainability of Societies." Ecological Economics 101 (May 2014): 90–102. https://doi.org/10.1016/j.ecolecon.2014.02.014.

(C) I strongly recommend adding a schematic diagram at the begging of the paper to show how POPEM interacts with CESM (e.g., variables, parameters, input/output, couplings).

(D) If POPEM + CESM is indeed the first model that calculates emissions at a local scale, as opposed to using global emissions projections, please emphasize that as the novel accomplishment of this research.

(E) Remove any parts of the manuscript that overlaps with Navarro et al. [2017], and instead refer to specific parts of that publication.

(F) Be more careful with the definitions of, and distinctions between, ESMs and IAMs. Navarro et al. [2017] write, for example: "[RCPs] provide simplified versions of human activities and processes, such as population density and economic development, from non-coupled Integrated Assessment Models (IAMs)." It is not true that IAMs are 'non-coupled'; they are indeed one-way coupled.

Then the authors write "researchers in the iESM Project (Collins et al. 2015) developed

a global integrated assessment model, the GCAM, to address human impact on climate dynamics, with special emphasis on the representation of the human earth system." GCAM was not developed in the iESM project, but has been in development since 1990s and is one of the leading IAMs. The rest of the description of the sentence is also incorrect. iESM couples land use and agriculture to ES via bidirectional feedbacks.

(G) In the last section of the manuscript (4), emphasize that dynamic models of various Human System components need to be developed and coupled to ESMs via bidirectional feedbacks in order to produce realistic projections and to capture counterintuitive and unexpected dynamics.

(H) Please go over your citations carefully and make sure that they appear at proper places. Also, the manuscript can benefit from additional important, relevant references. (The bibliography of Motesharrei et al. [2016] could be helpful for this manuscript.)

————————————————

End of the review of the manuscript "Improving the representation of anthropogenic CO2 emissions in climate models: a new parameterization for the Community Earth System Model (CESM)" by Andrés Navarro, Raúl Moreno, and Francisco J. Tapiador, Earth System Dynamics.

Submitted by Safa Motesharrei on 2018 MAY 05.

---

## Author Comment (AC1) · 1 Jun 2018

**Response to referee #1**

*Referee #1: The paper is very interesting, novel and merits immediate publication. The approach is that of a 'proof of concept', but the idea behind the research is extremely interesting and worth of attention by the community.*

**Reply:** Thank you very much. We really appreciate your comments and suggestions.

*Referee #1:*
However, I believe the authors must touch upon several topics in order to improve the paper. Specifically:
*Last part of section 3.1 needs further explanation. Please expand the section and provide more information about potential applications. I think that is an important part of the paper (probably the most important part), and it is a pity that the authors give just such a swift account of the topic.*

**Reply:** Thanks indeed. This is an important point that we didn't explain in full in the first version of the manuscript**.** We added two paragraphs now explaining the potential applications.

The text now reads:

> Potential applications of POPEM include not only sensitivity analyses of local $CO_2$ emissions policies, but also the added feature of performing tests for 'what-if' scenarios. One interesting example would be the climate response under the hypothesis that China and India –the most populated countries in the world- reach US $CO_2$ per capita emissions rates. Another 'what-if' scenario would be the climate response of an increasingly urbanized world. In both cases, POPEM provides a flexible framework for testing the alternative hypotheses.
>
> The realism of the ESM will be enhanced with a fully-coupled system. Such a fully-fledged ESM will include bidirectional feedback between POPEM and CESM to evaluate the effects of climate change on population dynamics and emissions.

*Referee #1: -Given the large number of papers using CESM I think more attention should be devoted to previous work using this model. Please add several references to show how CESM has been used, including merits, shortcomings and the like.*

**Reply:** We added two paragraphs in section 2.1. They include references from different topics.

The new paragraphs read:

*CESM –formerly the Community Climate System Model (CCSM)- was conceived as a coupled atmospheric-oceanic circulation model (Boville and Gent, 1998; Collins et al., 2006; Gent et al., 2011; Hurrell et al., 2013; Williamson, 1983). Since the release of the first version, CESM has evolved into a complex Earth System Model now used in different fields. This includes research into atmospheric (Bacmeister et al., 2014; Liu et al., 2012; Yuan et al., 2013), biogeochemical (Lehner et al., 2015; Nevison et al., 2016; Val Martin et al., 2014), and human-induced processes (Huang and Ullrich, 2016; Levis et al., 2012; Oleson et al., 2011), as well as others. The core code of CESM has also been utilized by various research centers for developing their own models (norESM, Bentsen, 2013; CMCC–CESM–NEMO, Fogli and Iovino, 2014; MIT IGSM-CAM, Monier et al., 2013). CESM has been used in many hundreds of peer-reviewed studies to better understand climate variability and climate change (Hurrell et al., 2013; Kay et al., 2015; Sanderson et al., 2017). Simulations performed with CESM have made a significant contribution to international assessments of climate, including those of the Intergovernmental Panel on Climate Change (IPCC) and the CMIP5/6 project (Coupled Model Intercomparison Project Phase 5/6) (Eyring et al., 2016; IPCC, 2014b; Taylor et al., 2012).*

*A major advantage of CESM over other ESMs is its availability. Some climate models are developed by scientific groups and access to the source code is limited. The CESM source code is free and available to download from the NCAR website. This approach helps improve the model by setting up a framework for collaborative research and makes the model fully auditable. CESM is a good example of a 'full confidence level' model, after Tapiador et al. (2017), where many 'avatars' of the code are routinely run in several independent research centers, and there is an entire community improving the model and reporting on issues and results. However, the model is not immune to bias. One important shortcoming is the poor representation of precipitation in terms of spatial structure, intensity, duration, and frequency (Dai, 2006; Tapiador et al., 2018; Trenberth et al., 2017, Trenberth et al., 2015). Another major bias is the anomalous warm surface temperature in coastal upwelling regions (Davey et al., 2001; Justin Small, 2015; Richter, 2015).*

*Referee #1: -The double ITCZ issue needs referencing. Who did first mention that? Without such reference it seems that that feature is a novel observation from the authors, which I think it is not.*

**Reply:** Sorry about that. We have added a citation.

The paragraph now reads:

*The first step in evaluating the new parameterization is to compare the outputs with a control simulation to make sure the new addition does not negatively*

*interact with the dynamical core or spoil the contributions of rest of the parameterizations. Figure 4 shows that this is not case with the POPEM parameterization, which does not negatively affect the outputs of precipitation and temperature. Rather, both variables are now closer to the observed data than they were in the control run, especially in terms of reducing the double ITCZ (Intertropical Convergence Zone), which artificially features in global models (Mechoso et al., 1995; for a recent analysis of double ITCZ in CMIP5 models see Oueslati and Bellon, 2015).*

**Referee #1:** *-The following sentence is confusing to me.˘The improvements of POPEM for the El Niño-4 area show that detailed, dynamical modeling of GHG emissions is important for more precisely quantifying precipitation in dry areas, which validates the main hypothesis of the paper. Please explain what do you mean by that.*

**Reply:** What we meant was that precipitation in dry areas is extremely important, since human activities and biota are highly dependent of it. Improving the representation of precipitation in models is thus crucial. The main hypothesis of the paper, namely that point-wise emissions can improve the modeling, is validated for the El Niño-4 area where we show that our model improves the representation of precipitation in the left tail of the distribution (cf. Figure 8). We have reworded the paragraph:

> *"The results for the El Niño-4 area show that detailed, grid-point emissions of GHG improves the quantification of precipitation in dry areas, in agreement with our hypothesis about the benefits of locally-distributed versus global mean forcings."*

**Referee #1:** *- Please, check the references to the figures in the text (Figs. 4 and 5)*

**Reply:** Amended now, thanks.

**Referee #1:** *- Please, check the place where you put the definition of some acronyms (e. g. ITCZ -you use it on page 7 and is defined in page 10-, SST -similar-).*

**Reply:** Sorry about that. Amended now.

**References**

Mechoso, C. R., Robertson, A. W., Barth, N., Davey, M. K., Delecluse, P., Gent, P. R., Ineson, S., Kirtman, B., Latif, M., Treut, H. Le, Nagai, T., Neelin, J. D., Philander, S. G. H., Polcher, J., Schopf, P. S., Stockdale, T., Suarez, M. J., Terray, L., Thual, O. and Tribbia, J. J.: The Seasonal Cycle over the Tropical Pacific in Coupled Ocean–Atmosphere General Circulation Models, Mon. Weather Rev., 123(9), 2825–2838, doi:10.1175/1520-0493(1995)123<2825:TSCOTT>2.0.CO;2, 1995.

Meehl, G. A. and Arblaster, J. M.: The Asian-Australian monsoon and El Nino-Southern Oscillation in the NCAR climate system model, J. Clim., 11(6), 1356–1385, doi:10.1175/1520-0442(1998)011<1356:TAAMAE>2.0.CO;2, 1998.

Oueslati, B. and Bellon, G.: The double ITCZ bias in CMIP5 models: interaction between SST, large-scale circulation and precipitation, Clim. Dyn., 44(3–4), 585–607, doi:10.1007/s00382-015-2468-6, 2015.

Terray, L.: Sensitivity of Climate Drift to Atmospheric Physical Parameterizations in a Coupled Ocean–Atmosphere General Circulation Model, J. Clim., 11(7), 1633–1658, doi:10.1175/1520-0442(1998)011<1633:SOCDTA>2.0.CO;2, 1998.

---

## Author Comment (AC2) · 1 Jun 2018

**Response to referee #3. Svetla Hristova-Veleva**

*Referee #3:*
*Overview and recommendations:*
*The paper addresses important questions regarding improving the performance of Earth System Models (ESM) – the important tools to study and understand the complexity of the Earth's climate. Improving these models is a major goal of the science community as they can be a very valuable tool in studying the response of the Earth's system to anthropogenic forcing, providing guidance to policy makers.*

**Reply:** Thanks.

*Referee #3:*
*In particular, the paper investigates the impact of a new parameterization of CO2 emissions that the authors have recently developed, called the POPEM module (POpulation Parameterization for Earth Models). POPEM presents an important advancement in the way CO2 emissions are modeled, as it accounts dynamically for the changing emissions. Like previous research, POPEM uses population data as proxies for emission. What is unique to this new parameterization, though, is that it models the evolution of the population while previous research has relied on historical data, hence not being dynamical, preventing them from making reliable predictions for the future emissions and the response of the climate system.*
*Using this new parameterization (POPEM) presents an important advancement and this makes the described research very valuable.*

**Reply:** Thanks.

*However, before going forward one have to evaluate the performance and assess the impact of the new parameterization. Indeed, this is the goal of this paper.*
*The paper begins by describing what is unique about POPEM.*
*It then validates the stand-alone performance of POPEM by comparing its predication over a past 63 (and 70) -year period to existing data. The comparison is done globally but also by several regions. This validation is done in two ways: by comparing forecasted to observed population growth rates; and by comparing the forecasted to observed emission rates. The results show that despite the difficulty of predicting non-linear trends in the growth of population and emissions, POPEM preforms quite well. These comparisons give credibility to the POPEM forecasts, hence to its use in forecasting future scenarios.*

*Next, the paper uses a coupled ESM, the Community ESM (CESM) to evaluate the impact of POPEM. The evaluation focusses on the impact of POPEM on two very important, and difficult to predict, parameters of the Earth's system - the precipitation and the sea surface temperature (SST). The evaluation is done in two ways:*
*- by comparing the results from a control run (using global CO2 concentration parameters that I believe are homogeneous – this needs clarification) to those*

*from POPEM. This choice of model setups highlights the value of POPEM as it predicts the population (and the emissions) in every grid point, showing the impact and the importance of the spatial variability.*
*- By comparing both control and POPEM forecasts to actual observations (over a 20-year period for precipitation and 50-year period for SST).*

*The paper finds that:*
*- The global predictions for both parameters compare to the observations in a very similar way for the CONTROL and the POPEM simulations. Hence, the more realistic POPEM parameterization "does no harm". This is an important test and conclusion because it is occasionally the case that including more realistic parameterizations might degrade the performance of the forecasts for certain parameters. This is because often the models are "tuned" to predicting some of the parameters, giving the right answer for the wrong reason, and impacting negatively the forecasting of the non-tuned parameters when the more realistic parameterizations are employed.*
*- More importantly, the paper finds that using POPEM results in regional differences between its forecasts and that of the control run. Comparison to observations seems to suggest the POPEM produces better regional distribution of the precipitation. This is a very important conclusion, in my view. It does not seem to be well highlighted in the paper summary.*

*Overall, the paper addresses a very important topic. The approach is sound and uses a very good modeling framework. There is a very extensive set of references. The paper is presented in a fluent and precise language. However, there are several places where the paper could be improved, as detailed below.*

*Because of all that, I propose the paper be accepted with minor revisions.*

**Reply**: Thanks for highlighting the main findings of the manuscript and for your detailed revision of the paper. Also, thanks for your suggestions and comments. We consider that they improve the global quality of the paper.

*Referee #3:*
*Title: The current title is: "Improving the representation of anthropogenic $CO_2$ emissions in climate models: a new parameterization for the Community Earth System Model (CESM)"*

*I would suggest a modification to read "Improving the representation of anthropogenic CO2 emissions in climate models: Impact of a new parameterization for the Community Earth System Model (CESM)"*

*The reason is that main goal of the paper is not to describe the new parameterization but to evaluate its performance and impact.*

**Reply:** Indeed, the suggested title describes more precisely the aim of the paper. Thanks. The title now reads:

> ***Improving the representation of anthropogenic $CO_2$ emissions in climate models: impact of a new parameterization for the Community Earth System Model (CESM).***

*Referee #3: Abstract*
*"The results show that it is indeed advantageous to model $CO_2$ emissions and pollutants directly at model grid points rather than using the forcing approach". Please, reword as it is not clear (at this point) what is this forcing approach.*

**Reply:** We rewrote the sentence to make the point clearer.

The text reads:

> *The results show that it is indeed advantageous to model $CO_2$ emissions and pollutants directly at model grid points rather than using the same mean value globally.*

*Referee #3:*
*Introduction:*
*The reader would benefit from a more detailed description of the existing approaches to modeling CO2 emissions. What I gather from the paper is the following: there are two basic approaches that models use to account for $CO_2$ forcing:*

*.    a) using globally homogenous forcing;*
*.    b) using non-homogenous, grid-point specific forcing. This one can be applied in several ways:*

*1.   using Representative Concentration Pathways (RCPs) that "are not fully-integrated socioeconomic parameterizations, but rather estimates for describing plausible trajectories of human climate change drivers .... They provide simplified accounts of human activities and processes, including population density and economic development, from non-coupled Integrated Assessment Models (IAMs;)" Question: are these parameters location- specific? This is what I am understanding.*

*2.   the proposed here POPEM model being integrated into a fully coupled model. This is similar to RCPs but: uses a coupled model; uses a dynamic model for the prediction of population and emissions.*

*.    c) Is my understanding correct???*

**Reply:** Our apologies. We did not make the point clear. It is the other way around: RCPs are used as a surrogate for point-wise estimates. We have clarified that in the revision of the paper [see next comments for more details]

*Referee #3:*
*.    d)  If so, I would suggest two possible modifications:*
*        1.  Use some wording or structure as what I've described above*
*        2.  Space-permitting, create either a small table or a flow diagram that shows these different levels of sophistication*

**Reply:** We have rewritten the two paragraphs to clarify the differences between RCPs and POPEM approaches. Thanks.

The amended paragraphs now read:

> *One of the fields most in need of development is the inclusion in global models of co-evolutionary dynamical interactions of the socioeconomic dimension into global models with other Earth system components (Nobre et al., 2010; Robinson et al., 2017; Sarofim and Reilly, 2011). Human activity was a major driver of change in the Earth System in the recent past (Alter et al., 2017; Barnett et al., 2008; Crutzen, 2002), and it now dominates the natural system (Ruth, et al. 2011). However, most global models use basic socioeconomic assumptions about the behavior of societies and are only unidirectionally linked to the biogeophysical part of the Earth system (Müller-Hansen et al., 2017; Smith et al., 2014). The standard way of introducing anthropogenic climate change into ESMs is through Representative Concentration Pathways (RCPs). These are consistent sets of projections involving only radiative forcing components (van Vuuren et al., 2011), but which represent a step forward from the scenario approach of the last decade (Moss et al., 2010; van Vuuren et al., 2014; van Vuuren and Carter, 2014). However, RCPs are not fully-integrated socioeconomic parameterizations but rather estimates for describing plausible trajectories of human climate change drivers (Moss et al., 2010; Vuuren et al., 2012). They provide simplified accounts of human activities and processes from one-way coupled Integrated Assessment Models (IAMs, Müller-Hansen et al., 2017).*

> *The use of RCPs is advantageous because they provide a set of pathways that serve to initialize climate models. However, two major problems remain within this approach. Firstly, human activities are not intrinsically embedded into the ESM, impeding sensitivity studies. Secondly, because of the weak coupling of IAMs, they cannot capture the sometimes counterintuitive bidirectional feedback and nonlinearity between the socioeconomic and natural subsystems (Motesharrei et al. 2016; Ruth et al. 2011). Good examples that illustrate the importance of including such bidirectional feedbacks feature in the HANDY model (Motesharrei et al. 2014) which has been used to analyze the key mechanisms behind societal collapses using the predator-prey model.*

> *The RCP approach has been used in climate models because of its low computational cost. However, advances in computational resources now allow to*

*parameterize human-Earth processes in a more detailed way, including the inclusion of population dynamics into the modeling, as in the POPEM (POpulation Parameterization for Earth Models) module (Navarro et al., 2017).*

**Referee #3:** *P. 2, lines 25-30 – It says: "Given the highly non-linear character of the processes involved, it is not unreasonable to assume that location is significant, and the spatial and time distribution of these emissions may affect global climate" – a bit unclear. Might be better to say ", it is not unreasonable to assume that specifying (or accounting for) geographical variability is significant"*

**Reply:** We modified the expression following your suggestion. The text now reads:

> *Given the highly non-linear character of the processes involved, it is not unreasonable to assume that accounting for geographical variability is significant, and the spatial and time distribution of these emissions may affect global climate (Alter et al., 2017; Grandey et al., 2016; Guo et al., 2013).*

**Referee #3:** *P. 3, lines 2-4: "The aim of this paper is to show that this grid point scale modeling of anthropogenic CO2 emissions (and other pollutants) represents an improvement, and that two important variables, namely global precipitation distribution and surface temperature, are not negatively affected by this more-detailed approach." While this is true I believe this is a rather weak statement regarding the benefits of using POPEM-type parameterization of emissions forecasting. I believe the authors are in a position to make a stronger statement,*
*namely: including the POPEM dynamical forecasting approach that accounts for the spatial and temporal variability of the emission sources, leads to better representation of the geographical variability of the precipitation.*

**Reply:** We rewrote the las part of the paragraph to include your suggestion.

The text now reads:

> *The aim of this paper is to show that this grid point scale modeling of anthropogenic $CO_2$ emissions (and other pollutants) represents an improvement over simpler approaches, and leads to better representation of the geographical variability of precipitation.*

**Referee #3:** *Space-permitting, I would suggest that the **Introduction** ends with a short description of the outline for the following presentation. Something like: " the following sections outline: the unique features of POPEM; the validation of the POPEM stand-alone performance; the framework for evaluating the impact of POPEM – incorporation into CESM and framework for testing; the comparison between a control run and a POPEM-specific one: evaluating the differences between the two; evaluating how each compares*

*to observations; discussions; summary and conclusions;" This would give the reader a clear structure of the paper to follow and will make it easier to highlight the contributions of the paper.*

**Reply**: Thanks for the suggestion. We added a new paragraph with a short description of the outline.

The new paragraph reads:

> *The paper is organized as follows: in section 2, we present the validation of the POPEM standalone mode and set the framework for evaluating the impact of POPEM parameterization –its incorporation into the CESM and the testing framework; in section 3, we compare the outputs of CONTROL and POPEM runs and see how they compare with observations. In the conclusion and future work section, we highlight the importance of the dynamical modeling of anthropogenic emissions at grid point scale to better represent the socioeconomic parameters in the CESM model and improve precipitation estimates.*

*Referee #3:*
*Section 2.2*
*currently there are sections 2.2 and 2.2.1 but not 2.2.2 or more. It seems that there is no need for 2.2.1. If there is no 2.2.2. I would suggest the following: "2.2 POPEM specifics and validation", followed by "2.2.1 POPEM parameterization model overview: Unique features" and "2.2.2 POPEM trend verification". Of course, this is just a suggestion.*

**Reply:** Thanks for the suggestion. We rewrite subsection titles and numbers to have a clearer structure.

Now, subsections titles are:
> *2.2 POPEM specifics and standalone validation*
> > *2.2.1 POPEM parameterization model overview*
> > *2.2.2 POPEM trend verification*

*Referee #3: P. 6, lines 8-9 – "Our control case used global CO2 concentration parameters (standard procedure in ESMs), while the POPEM case used geographically-distributed CO2 emissions data" - is the control using homogeneous CO2 concentrations? I am pretty sure this is the case but it might be better to say it this way.*

**Reply:** [already discussed above] We have replaced the word **'global'** with the word **'homogeneous'** to make it clearer.

Text now reads:

*Our control case used **homogeneous** $CO_2$ concentration parameters (standard procedure in ESMs), while the POPEM case used geographically-distributed $CO_2$ emissions data.*

*Referee #3:*
*Section 3.1*
*P.7, line 23 – it appears that figures 6C, 6D, 8C and 8D are referenced before figures 4 and 5 (and the figure 8 is referenced before Fig.7). This should not be the case. The figures should be referenced in order. However, it seems that this is because the current order of the discussions here might need to be modified. Below is what I mean.*
*a) Maybe the order should be: 1. Test for "no harm" – figures 6C-6D and 8C-8D show that. 2. Compare the CONTROL to the POPEM simulations to see where exactly they differ. 3. Compare both the CONTROL and the POPEM CESM simulations to the observations, looking at regional distributions. The comparison in steps 2 and 3 brings up the impact of the POPEM geographically-aware CO2 emissions on the geographical distribution of the precipitation, highlighting the positive impact POPEM has (especially in step3).*
*b) Steps 2 and 3 could be switched – depending on what the authors think.*
*c) I want to point out that the proposed change in the order of the presentation is just a suggestion for the authors to consider.*

**Reply:** Thanks for the suggestion. We have restructured the order of the figures to make it clear.

*Referee #3: P.8, lines 2-3: "It is clear from the figure that POPEM does alter the spatial pattern of precipitation and exerts a definite effect on the climate pattern, as the module reduces the otherwise exaggerated ITCZ precipitation in the Southern Hemisphere (South East Asia and Australia)." Do you have a reference that it was exaggerated?? If so, then this is a very strong point that needs to be emphasized. Also, do you mean Fig. 4 or Fig. 5? Please, specify.*

**Reply**: The double ITCZ bias is a persistent problem in most climate models. It has been reported by several authors (Mechoso, 1995; Terray, 1997; Lin 2007) and the causes of this bias are still unclear (Li and Xie, 2014). In the Southern Hemisphere, climate models produce an excess of precipitation in the band 10S-15S when compared with satellite observations (Hwang and Frierson, 2012). We have added a few citations to highlight the importance of this issue.

Additionally, we made a new figure (Figure 9) to clarify the improvements of POPEM in the double ITCZ bias [see the next reply].

The paragraph now reads:

*It is clear from Figures 5A and 6A that POPEM does alter the spatial pattern of precipitation and exerts a definite effect on the climate pattern, as the module*

*reduces the otherwise exaggerated ITCZ precipitation in the Southern Hemisphere reported by several authors (Hwang and Frierson, 2013; Lin and Xie 2014).*

**Referee #3:** *P. 8, lines 7-8: "There are also important differences in precipitation in the 30N-30S band. Here POPEM reduces model bias, especially in the Southern Hemisphere and on the Tibetan Plateau." How do we know that the model bias is reduced?*

**Reply**: We have now explained this point in full in the section 3.2 and also made a new figure to clarify the point (Figure 9).

Figure 9A shows monthly precipitation for the area affected by the double ITCZ bias in the Southern Hemisphere (20S-0, 80E-100W). It is clear from this figure that POPEM yields more realistic representation of precipitation especially in the driest months (June-October). Figures 9B and 9C show the annual cycle of rainfall over the Australia Top End region and over the Tibetan Plateau, respectively. In both instances there is a usual bias in the original CESM. We have noted that despite POPEM obtaining slightly better results, both CONTROL and POPEM still have difficulties to estimate the precipitation of the rainiest months.

The paragraph now reads:

> *Another important benefit of POPEM is the reduction of the double ITCZ bias in the Southern Hemisphere. Although a small change can be inferred from Figure 7A-B, the improvement is buried in the annual mean precipitation maps. Figure 9A shows that the POPEM results are closer to observations of the intra-annual variability of precipitation, especially for the driest months (June-October).*

[Figure]

*Figure 9: Monthly precipitation (1980-1999) based on GPCP, CTRL and POPEM for three of the regions with important biases in CESM. (A) shows precipitation for the area affected by the double-ITCZ bias in the Southern Hemisphere (20S-0, 80E-100W); (B) for Australia Top End (30S-10S, 128E-140E); and (C) for the Tibetan Plateau (22N-32N, 78W-92W). The black line represents observations (GPCP), the blue line is the CONTROL case, and the red line is the POPEM case. Units are in mm/day. The arrow indicates the improvement of the POPEM model.*

*The figure also shows slight improvements for another two typical biases seen in CESM, namely the excess precipitation in the Tibetan Plateau (Chen and Frauenfeld, 2014; Su et al., 2013; Figure 9C) and the bias in some areas affected by the Asian-Australian monsoon (AAM), such as the Australia Top End (Meehl and Arblaster, 1998; Meehl et al. 2012; Figure 9B).*

**Referee #3:** *P. 8, line 9-10: "On the other hand, POPEM departs from the control simulation in the Asia-Pacific region between 10N-10S." Is that good or bad? How do we know?*

**Reply**: If we zoom-in on figure 6A (map: CONTROL minus POPEM) it can be seen that POPEM produces more precipitation than CONTROL. That means that the model reinforces the double ITCZ bias in this area, which is not good. We have noted that in the paper.

The text reads now:
> *On the other hand, POPEM departs from the control simulation in the Asia-Pacific region between 10N-10S. This result reinforces the double ITCZ bias in this area.*

**Referee #3:** *P. 8, line 31 – "(Q1 and Q3 remain between ± 0.4 mm/day)." Please, define Q1 and Q3.*

**Reply:** Q1 and Q3 mean Quartile 1 and Quartile 3. We now write down the word in full to avoid possible confusion.

The line now reads:
> *(The first and the third quartiles of the distribution remain between $\pm 0.4$ mm/day)*

**References**

Hwang, Y.-T. and Frierson, D. M. W.: Link between the double-Intertropical Convergence Zone problem and cloud biases over the Southern Ocean., Proc. Natl. Acad. Sci. U. S. A., 110(13), 4935–40, doi:10.1073/pnas.1213302110, 2013.

Li, G. and Xie, S. P.: Tropical biases in CMIP5 multimodel ensemble: The excessive equatorial pacific cold tongue and double ITCZ problems, J. Clim., 27(4), 1765–1780, doi:10.1175/JCLI-D-13-00337.1, 2014.

Lin, J. L.: The double-ITCZ problem in IPCC AR4 coupled GCMs: Ocean-atmosphere feedback analysis, J. Clim., 20(18), 4497–4525, doi:10.1175/JCLI4272.1, 2007.

Meehl, G. A. and Arblaster, J. M.: The Asian-Australian monsoon and El Nino-Southern Oscillation in the NCAR climate system model, J. Clim., 11(6), 1356–1385, doi:10.1175/1520-0442(1998)011<1356:TAAMAE>2.0.CO;2, 1998.

Mechoso, C. R., Robertson, A. W., Barth, N., Davey, M. K., Delecluse, P., Gent, P. R., Ineson, S., Kirtman, B., Latif, M., Treut, H. Le, Nagai, T., Neelin, J. D., Philander, S. G. H., Polcher, J., Schopf, P. S., Stockdale, T., Suarez, M. J., Terray, L., Thual, O. and Tribbia, J. J.: The Seasonal Cycle over the Tropical Pacific in Coupled Ocean–Atmosphere General Circulation Models, Mon. Weather Rev., 123(9), 2825–2838, doi:10.1175/1520-0493(1995)123<2825:TSCOTT>2.0.CO;2, 1995.

Meehl, G. A. and Arblaster, J. M.: The Asian-Australian monsoon and El Nino-Southern Oscillation in the NCAR climate system model, J. Clim., 11(6), 1356–1385, doi:10.1175/1520-0442(1998)011<1356:TAAMAE>2.0.CO;2, 1998.

Meehl, G. A., Arblaster, J. M., Caron, J. M., Annamalai, H., Jochum, M., Chakraborty, A. and Murtugudde, R.: Monsoon regimes and processes in CCSM4. Part I: The Asian-Australian monsoon, J. Clim., 25(8), 2583–2608, doi:10.1175/JCLI-D-11-00184.1, 2012.

Terray, L.: Sensitivity of Climate Drift to Atmospheric Physical Parameterizations in a Coupled Ocean–Atmosphere General Circulation Model, J. Clim., 11(7), 1633–1658, doi:10.1175/1520-0442(1998)011<1633:SOCDTA>2.0.CO;2, 1998.

---

## Author Comment (AC3) · 1 Jun 2018

**Response to referee #4.**

*Referee #4: The authors developed a novel POPEM parameterization and applied it to CESM to enhance the realism of global climate modeling by improving the direct representation of human activities and climate. They argued that modeling CO2 emissions and pollutants directly at model grid points is a better approach. As such, their new approach will help understand the potential effects of localized pollutant emissions on long-term global climate statistics, thus assisting adaptation and mitigation policies. The topic is interesting and the approach is provoking.*

**Reply:** Thank you for your positive feedback.

*Referee #4: However, I am not quite convinced by the validation part (Part 3.2). I therefore recommend major revision.*

**Reply:** We followed your recommendations. We have expanded section 3.2 and added a new subsection; *3.3 Validation against ESPI and ONI indices.* Please, see following comments for a detailed revision of the updates. Hope the changes can solve your concerns.

*Referee #4: First, I cannot find a remarkable improvement using POPEM based on the comparison of precipitation and temperature biases. There are some differences between POPEM and CONTROL but these differences are buried in the large biases in either set.*

**Reply:** We have made clearer in the paper that we do not claim to solve the problem of homogenous emissions versus point-wise estimates. We did not state that our contribution produces a remarkable improvement. What we have achieved by now is far more modest: we have shown that including our more-realistic forcings preserves the model ability to produce realistic fields. Nonetheless, some improvements can be seen (we have included additional figures to illustrate the improvements). We agree that the improvements are limited, but given the small model sensitivity to this forcing (the logic of RCP85 is to somehow 'exaggerate' the emissions to increase the signal), one cannot expect major changes. In other words, the actual signal is too faint to be affected by a more realistic emission pattern. Indeed, the reason for having a distributed method is to be able to evaluate 'what-if' scenarios (i.e. what happens if China cuts off emissions, or the like). We have added a paragraph at the end of the section 3.1 to explain why the approach is valuable in spite of the marginal improvements compared with validation data.

As referee #5 says, we also believe that the use of local population projections to project emissions at each grid point is novel, and is advantageous to the current practice of using global emissions projections to drive ESMs.

The added paragraph reads:

*Potential applications of POPEM include not only sensitivity analyses of local $CO_2$ emissions policies, but also the added feature of performing tests for 'what-if' scenarios. One interesting example would be the climate response under the hypothesis that China and India –the most populated countries in the world- reach US $CO_2$ per capita emissions rates. Another 'what-if' scenario would be the climate response of an increasingly urbanized world. In both cases, POPEM provides a flexible framework for testing the alternative hypotheses.*

**Referee #4:** *It is true that observations have uncertainties and a new parameterization does not have to improve the model performance in every aspect. Nevertheless, could the authors show some improvements more robust than the current ones (precipitation and temperature) for validation? Maybe TOA radiation balance, ENSO index, Arctic sea ice, etc?*

**Reply:** We agree that the analysis of Artic sea ice response would be a good addition. Unfortunately, sea ice was not a focus of our research when we ran the simulations and now it is too late to do so. Same about TOA. However, in order to satisfy this requirement, we have included two additional validation metrics using two ENSO indices: namely the ENSO Precipitation Index (ESPI) and the Oceanic el Niño Index (ONI).

We have chosen the ESPI index, which estimates the gradient of the anomalies across the Pacific basin (Curtis and Adler, 2000). It compares well with SST-and pressure-based indices and is widely used by the scientific community (Figure 13 now). The Oceanic el Niño Index is a SST index developed by NOAA as a principal measure for monitoring, assessing and predicting ENSO (Kouski and Higgins, 2007).

We have made two new figures and added a table: Figure 13 for ESPI index, El Niño (EI) and La Niña (LI), and Table 1 and Figure 14 for ONI.

The new section reads as follows:

**3.3 Validation against ESPI and ONI indices**

[revised manuscript text omitted]

*Referee #4: Actually, I am somewhat interested in the Arctic sea change. It is known that climate models (like CESM CONTRL) cannot capture a rapid observed decline of Arctic sea ice during recent decades. In Fig. 5(B), POPEM is colder than CONTROL over the Barents Sea area. Will this mean that Arctic sea ice decline in POPEM is even slower than that in CONTROL?*

**Reply:** It's true that the POPEM parameterization produces colder temperatures in that area and that might reinforce the bias of a slower Artic sea ice decline. Unfortunately, we can't contrast this hypothesis because we did not keep the sea ice outputs for our simulations. Sorry about that.

The bias is less evident when confronted with GISTEMP annual mean anomalies for that area. It is seen from the Figure 11 (top) that CONTROL and POPEM cases have a similar margin error. In other words, the original CESM model is not really good in capturing this feature. Our approach slightly improves the situation in some cases (Bering Sea from 1975 to 1990, Figure 11 (middle)) but we cannot expect a major overall improvement.

We have added a paragraph and a figure to clarify this point.

The text now reads:

*The bias is also reproduced when compared with temperature anomalies for a specific region. Thus, for instance, CESM gives poor scores in the Barents Sea area (Figure 11; top) while POPEM obtains better results for the Bering Sea, especially in the Russian part (Figure 11; middle). Here, POPEM gives more realistic values for the period 1970-1998 but, even with the improvement, the model still overestimates the temperature anomaly.*

[Figure]

**Figure 11**: *A comparison of the annual mean surface temperature anomaly between GISTEMP, CONTROL and POPEM from 1950 to 1999. (Top) represents the Barents Sea (68N-80N, 19E-68E); (middle) Russian part of the Bering Sea (50N-65N, 150E-180E); and (bottom) American part of the Bering Sea (50N-75N, 140W-180W). The black line represents observational data (GISTEMP), the blue line is the CONTROL case, and the red is the POPEM case. Anomaly was referenced to 1951-1980 period.*

We also calculated the temperature anomalies with monthly data (attached as a supplementary material). However, the noise is high and it is difficult to distinguish any clear pattern other than the consistency between the series. Only in Figure EXT2(top) we see that POPEM more frequently yields extreme values.

[Figure]

Figure EXT2: The same as Figure 11 but using monthly means.

*Referee #4: Besides, to be consistent with GPCP, the authors may want to use a globally (land+ocean) covered temperature dataset GISTEMP (https://data.giss.nasa.gov/gistemp/) to examine temperature bias.*

**Reply:** Thanks for the suggestion. As you seen in the previous comment we included GISTEMP in several figures and also made a brief description of the source in section 2.4.3.

The new subsection reads:

> ### 2.4.3 GISTEMP data set
> *NASA's GISTEMP (GISS Surface Temperature Analysis) is a global surface temperature change dataset (Hansen and Lebedeff, 1987; see Hansen et al. 2010 for an updated version). It combines land and ocean surface temperatures to create monthly temperature anomalies at $2^o$ x $2^o$ degrees of spatial resolution. The use of anomalies reduces the estimation error in those places with incomplete spatial and temporal coverage (Hansen and Lebedeff, 1987). The anomalies are calculated over a fixed base period (1951-1980) that makes the anomalies consistent over long periods of time.*
> *The first version was originally conceived only for land areas (Hansen and Lebedeff, 1987) but in 1996 marine surface temperatures were added (Hansen et al., 1996). The updated version of GISTEMP includes satellite-observed nightlights to identify stations located in extreme darkness and adjust temperature trends of urban stations for non-climatic factors (Hansen et al. 2010). Just like CRUTS, GISTEMP is commonly used to validate climate models because of its coverage and confidence levels (Baker and Taylor, 2016; Brown et al., 2015; Neely et al., 2016, Peng et al., 2015).*

Additionally, we used GISTEMP to analyze temperature anomalies for regional (previous comment; Figure 11) and global scales (Figure 12).

The results of Figure 12 were discussed in the section 3.2 of the manuscript:

The new paragraph reads as follows:

> *If we focus on global temperature anomalies, CESM simulations are able to reproduce the progressive increase in the temperature anomaly (Figure 12; top). However, the CONTROL case simulates a sharp drop at the end of the period (1990-1999), while POPEM portrays this change as smooth, in agreement with the observations.*

[Figure]

**Figure 12:** A comparison of the global annual mean surface temperature anomaly between GISTEMP, CONTROL, and POPEM from 1950 to 1999. (Top) global; (middle) land; and (bottom) ocean. The black line represents observational data (GISTEMP), the blue line is the CONTROL case, and the red is the POPEM case. Anomaly was referenced to 1951-1980 period.

*The differences between CONTROL and POPEM are better demonstrated when comparing land and ocean separately (Figure 12; middle and bottom). While the temperature anomalies for land are quite similar in both cases, POPEM provides a better representation of the ocean tendency from 1992 onwards, and that translates to an overall improvement (Figure 12, top).*

We also computed monthly mean temperature anomalies. However, is difficult to appreciate the differences between models, especially for cases A and B. The figure is therefore included as a supplementary material.

[Figure]

*Figure EXT3: Same as Figure 12 but for monthly mean temperature anomaly. The main tendency is consistent albeit differences exists. Thus for instance the POPEM model clearly improves over CONTROL from 1992 onwards.*

---

## Author Comment (AC4) · 1 Jun 2018

**Response to referee #5. S. Motesharrei**

*Referee #5:*
*Review of the manuscript "Improving the representation of anthropogenic $CO_2$ emissions in climate models: a new parameterization for the Community Earth System Model (CESM)" by Andrés Navarro, Raúl Moreno, and Francisco J. Tapiador, submitted to the Journal Earth System Dynamics, European Geosciences Union (EGU).*
*Decision:*
*Because of the importance of the topic, I would recommend the publication of this manuscript after major revisions in the presentation of the work as described below.*

**Reply**: Many thanks for your positive feedback. Please, see following comments for a detailed revision of the updates.

*Referee #5:*
*General comments:*
*The authors acknowledge (but not completely clearly) a major shortcoming of the Earth System Models (ESMs) and Integrated Assessment Models (IAMs).*
*Even though the Human System has become the dominant driver of most components of the Earth System since about 1750, and especially since about 1950, IAMs use independent, exogenous projections of the Human System (HS) variables in order to drive ESMs to create future projections. Not including essential bidirectional feedbacks between ES and HS can lead to missing important dynamics that is critical to the sustainably of our planet and people. This problem is discussed in detail in the "Modeling Sustainability" paper by Motesharrei et al. [2016]:*

*Motesharrei, Safa, Jorge Rivas, Eugenia Kalnay, Ghassem R. Asrar, Antonio J. Busalacchi, Robert F. Cahalan, Mark A. Cane, et al. "Modeling Sustainability: Population, Inequality, Consumption, and Bidirectional Coupling of the Earth and Human Systems." National Science Review 3, no. 4 (December 11, 2016): 470–494. https://doi.org/10.1093/nsr/nww081.*

**Reply**: Many thanks for giving us the opportunity to expand this point in the paper. Also, thanks for the reference, which reinforce our point. We have used it to expand the issue in the revised version of the manuscript.

We extended the third paragraph of the introduction section to explain the point in details. The text now reads:

> *One of the fields most in need of development is the inclusion in global models of co-evolutionary dynamical interactions of the socioeconomic dimension into global models with other Earth system components (Nobre et al., 2010; Robinson et al., 2017; Sarofim and Reilly, 2011). Human activity was a major driver of change in the Earth System in the recent past (Alter et al., 2017; Barnett et al., 2008; Crutzen, 2002), and it now dominates the natural system (Ruth, et al.*

*2011). However, most global models use basic socioeconomic assumptions about the behavior of societies and are only unidirectionally linked to the biogeophysical part of the Earth system (Müller-Hansen et al., 2017; Smith et al., 2014). The standard way of introducing anthropogenic climate change into ESMs is through Representative Concentration Pathways (RCPs). These are consistent sets of projections involving only radiative forcing components (van Vuuren et al., 2011), but which represent a step forward from the scenario approach of the last decade (Moss et al., 2010; van Vuuren et al., 2014; van Vuuren and Carter, 2014). However, RCPs are not fully-integrated socioeconomic parameterizations but rather estimates for describing plausible trajectories of human climate change drivers (Moss et al., 2010; Vuuren et al., 2012). They provide simplified accounts of human activities and processes from one-way coupled Integrated Assessment Models (IAMs, Müller-Hansen et al., 2017).*

*The use of RCPs is advantageous because they provide a set of pathways that serve to initialize climate models. However, two major problems remain within this approach. Firstly, human activities are not intrinsically embedded into the ESM, impeding sensitivity studies. Secondly, because of the weak coupling of IAMs, they cannot capture the sometimes counterintuitive bidirectional feedback and nonlinearity between the socioeconomic and natural subsystems (Motesharrei et al. 2016; Ruth et al. 2011). Good examples that illustrate the importance of including such bidirectional feedbacks feature in the HANDY model (Motesharrei et al. 2014) which has been used to analyze the key mechanisms behind societal collapses using the predator-prey model.*

*The RCP approach has been used in climate models because of its low computational cost. However, advances in computational resources now allow to parameterize human-Earth processes in a more detailed way, including the inclusion of population dynamics into the modeling, as in the POPEM (POpulation Parameterization for Earth Models) module (Navarro et al., 2017).*

*Referee #5: The manuscript is closely related to a recently published paper by the same team of authors (and, unfortunately, there is much overlap with that already published work):*
*Navarro, Andrés, Raúl Moreno, Alfonso Jiménez-Alcázar, and Francisco J. Tapiador. "Coupling Population Dynamics with Earth System Models: The POPEM Model." Environmental Science and Pollution Research, September 16, 2017, 1–12. https://doi.org/10.1007/s11356-017-0127-7.*

**Reply**: Actually, there are major differences with that paper. Navarro et al. (2017) described in detail the **demographic** part of POPEM. In that paper, we were focused on the explanation and validation of the demographic and emission parts at global scale. In contrast, the current paper deals with the coupling of that demographic model with an **Earth System Model**, and compare the model outputs with observational data. That is a completely different history.

Also, the original emissions modeling module has been improved. We included a new figure (EXT1) in the supplementary material to show that. It looks:

[Figure]

**EXT1:** Comparison of the historical global $CO_2$ emission estimates for the years 1950–2012. The black line shows the estimates given using POPEM v1, red indicates POPEM v2, and purple indicates CDIAC estimates. Values are given in million of metric tonnes.

We have now limited the potential overlaps to the minimum required for the paper to be self-contained. We have rewritten parts of that section and added a new paragraph in the *2.2.1 POPEM parameterization model overview* subsection to clarify the novelties between successive POPEM versions and how the changes affect the emission estimates and the coupling with the model.

The new paragraph reads:

> *The demographic/emissions module presented here is an updated version of the demographic module explained in Navarro et. al (2017). The differences between the versions are minimal. They involve better approximation of emissions in highly polluting regions with poor population data, such as China; a better estimate for coastal zones and country limits; and a change in the model time step for more efficient coupling with CESM. The inclusion of these changes results in more accurate emissions estimates when compared with inventories than the previous version did. However, the model is not immune to bias. The most important limit is the degradation of the model outputs when there is increased spatial resolution –resolution of $0.25^o$ and higher–.*

*Referee #5: These two papers take a step toward including at least parts of the Human System (human population and emissions) explicitly in the ESMs, however, the somewhat in- accurate presentation of the work (and occasional over-statements) may lead to readers' confusion about the extent and novelty of this work. During my initial*

*reading of the manuscript, I was very impressed by the model and thought that it is a bidirectionally coupled Human System + Earth System Model. (It seems Anonymous Referee 3 has this same impression.) But upon further reading of the manuscript as well as Navarro et al. [2017], I realized that POPEM is essentially a demographic projection model (although it uses dynamic variables for age cohorts) that is used to drive CESM.*

**Reply**: Sorry if the description of POPEM in the first version of the manuscript was unclear. We have now amended the explanation to avoid the confusion [cf. reply to section (A)].

*Referee #5:* By contrast, I believe the use of local population projections to project emissions at each grid point is novel, and is advantageous to the current practice of using global emissions projections to drive ESMs.

**Reply**: Thank you for noting this. We believe that this is the central idea of the paper.

*Referee #5:*
*Suggested Revisions:*
*The other three referees already provide many helpful, important suggestions to improve the manuscript. Here, I outline some additional suggestions to help accurately present the model, its value for the Earth System modeling community, and possible future steps that needs to be taken by the modeling community to make the projections of the "Earth–Human System Models" more realistic.*

**Reply**: Many thanks for your valuable comments to improve the model.

*Referee #5:* I do not ask for any changes to model, since such changes would require major effort and could be implemented in future versions.

**Reply**: Thanks for your understanding and consideration; really appreciate it.

*Referee #5:* (A) Clarify that POPEM is, after all, a demographic projection model. It is true that its 18 age cohorts are dynamic variables, however, they still change based on exogenous fertility and mortality rates

**Reply**: Sorry if that was not clear in the first version of the manuscript. We have now extended the first paragraph of *2.2.1 POPEM parameterization model overview* to make it clear. We also redesigned Figure 1 highlighting now the external parameters.

The paragraph now reads:

> *The POPEM module is a demographic projection model coded in FORTRAN that is intended to estimate monthly fossil fuel $CO_2$ emissions at model grid point scale using population as the input. Due to a lack of actual GHG measurements at*

*appropriate spatial and temporal scales, it is necessary to use a proxy. For this, POPEM employs population, the evolution of which is modeled using external parameters that feed the module.*

***Referee #5:*** *(POPEM does not model Migration, which has become a major driver of population change, especially recently.)*

**Reply**: Modeling migration flows is an important point that we have taken into account since the very beginning of this project because it is a key element of population change –present and future-. However, there are several restrictions to accuracy estimate migration flows for historical populations at grid cell scale. Firstly, there are two different types of fluxes –short and long distance migrations- that have to be modeled in different ways (Lenormand et al. 2016). Secondly, we must quantify the entering and the exiting population for each cell using a probability rate of migration that is difficult to estimate with the limited migration data (Navarro et al. 2017). Thirdly, it is difficult – but not impossible- to validate a highly-detailed migration model with limited availability of migration data. Fourthly, the computational cost rises dramatically (e.g. *4 types of migration fluxes* x *number of cells* x *age-group* x *number of timesteps*). Consequently, these sources of uncertainties are greater than the benefits for the period of time and the spatial resolution used in this work.

***Referee #5:*** *These rates are projected into the future using statistical methods such as in the UN Population Projections. Therefore, the projections using POPEM could not be much different from traditional demographic projections, as can be seen from comparisons of POPEM to UN projections in Navarro et al. [2017]. I believe indeed POPEM cannot properly capture demographic change details for some regions and for certain age cohorts. Therefore, the value-added from this 'dynamic' population model is limited, at least from a demographic perspective.*

**Reply**: We assume that there is room for improvement in the demographic part of the model and it is an important point that we have to develop in the future versions of POPEM. However, the time period that we used here (1950-2000) and the actual spatial resolution offered by POPEM ($1^{o}$ x $1^{o}$) make model outputs less sensible to the referred biases. We have nonetheless clarified the limitations of the approach in the revised version. [see above the reworked text]

***Referee #5:*** *(B) Because ES and other components of the HS do not feedback onto the demographic variables in POPEM, POPEM will not be able to capture non-trivial dynamics that can arise due to such bidirectional feedbacks [Motesharrei et al., 2016]. For basic examples of how these bidirectional feedbacks (in a minimal model) can lead to surprising behavior, see:*
*Motesharrei, Safa, Jorge Rivas, and Eugenia Kalnay. "Human and Nature Dynamics (HANDY): Modeling Inequality and Use of Resources in the Collapse or Sustainability of*

*Societies." Ecological Economics 101 (May 2014): 90–102. https://doi.org/10.1016/j.ecolecon.2014.02.014.*

**Reply**: Firstly, thank you for this crucial reference. We considered that the citation of this work in the first part of the manuscript clarifies how important are the Human-Earth interactions and their feedbacks for models.

Secondly, we agree with you that bidirectional feedbacks between ES and HS are essential to make ESMs more accurate and realistic. The work presented here is just the first step in that direction.

[See the second comment in the discussion to check how we have expanded this point in the revised version of the manuscript.]

*Referee #5: (C) I strongly recommend adding a schematic diagram at the begging of the paper to show how POPEM interacts with CESM (e.g., variables, parameters, input/output, couplings).*

**Reply**: Thanks. We have reworked Figure 1 following your recommendations.

Figure 1 now looks:

[Figure]

**Figure 1:** Conceptual schema of the POPEM module coupled with the CAM5 atmosphere module. POPEM requires three input data sets to compute emissions (black dashed rectangles): initial population distribution; demographic parameters (age structure, death, and birth rates); and per capita emission rates by country. POPEM provides a 3D array (time, latitude, longitude) with emissions that are read by the *CO2_cycle* module and passed to the *atm_comp_mct* module which computes the total amount of $CO_2$ in the atmosphere.

*Referee #5: (D) If POPEM + CESM is indeed the* first *model that calculates emissions at a local scale, as opposed to using global emissions projections, please emphasize that as the novel accomplishment of this research*

**Reply**: Thanks for the suggestion. We added two sentences in the las part of the first paragraph (section 2.2.1).

The extended version now reads:

> *[...]The idea of using population as proxy is not new, and population density has previously been used to downscale national $CO_2$ emissions (Andres et al., 1996, 2016). However, these inventories were not dynamical, but instead tied to historical data so it is not possible to use them either to estimate future changes in emissions, or coupled with other components of the model. This change represents an important advance in the way emissions are computed. Thus, POPEM uses a bottom-up approach, where emissions are calculated at cell level on the basis of population dynamics, while global inventories use a top-down approach, which is less flexible when coupled with other components of the ESM.*

**Referee #5:** *(E) Remove any parts of the manuscript that overlaps with Navarro et al. [2017], and instead refer to specific parts of that publication.*

**Reply**: We have removed some overlapping text and referred to Navarro et al. 2017. However, there are some elements that it is important to keep in the manuscript for the reasons mentioned at the beginning of this discussion (see reply to third comment). Hope you find the reasons compelling enough to justify our choice.

**Referee #5:** *(F) Be more careful with the definitions of, and distinctions between, ESMs and IAMs. Navarro et al. [2017] write, for example: "[RCPs] provide simplified versions of human activities and processes, such as population density and economic development, from non-coupled Integrated Assessment Models (IAMs)." It is not true that IAMs are 'non- coupled'; they are indeed one-way coupled.*

**Reply**: Sorry about that. What we wanted to say here was 'one-way coupled'.

**Referee #5:** *Then the authors write "researchers in the iESM Project (Collins et al. 2015) developed a global integrated assessment model, the GCAM, to address human impact on climate dynamics, with special emphasis on the representation of the human earth system." GCAM was not developed in the iESM project, but has been in development since 1990s and is one of the leading IAMs. The rest of the description of the sentence is also incorrect. iESM couples land use and agriculture to ES via bidirectional feedbacks.*

**Reply**: Sorry about that. Perhaps we should have described more precisely that GCAM is the IAM used by the iESM model in that paper. We take note of that for the future.

**Referee #5:** *(G) In the last section of the manuscript (4), emphasize that dynamic models of various Human System components need to be developed and coupled to ESMs via*

*bidirectional feedbacks in order to produce realistic projections and to capture counterintuitive and unexpected dynamics.*

**Reply**: Thanks for the suggestion. We added a concluding paragraph in the manuscript.

The new paragraph reads:

> *Although the version of POPEM presented here is already functional, this work is intended to be just the first step in fully coupling socioeconomic dynamics with ESMs. This will include bidirectional feedback between Human and Earth systems and the simulation of societal processes based on the internal dynamics of the model instead of using external sources to make the projections. Only within a coupled global Human-Earth system framework can we produce more realistic representations of the Earth system capturing much of the counterintuitive feedback that is missing from current models (Motesharrei et al. 2016). The success of this approach will depend on the ability of scientists from different research fields to work in an interdisciplinary framework of continuous collaboration.*

*Referee #5: (H) Please go over your citations carefully and make sure that they appear at proper places. Also, the manuscript can benefit from additional important, relevant references. (The bibliography of Motesharrei et al. [2016] could be helpful for this manuscript.)*

**Reply**: Thank you for the advice and the reference. That excellent review helped us to find new relevant references, such as the previous work done by Matthias Ruth, Eugenia Kalnay and Jorge Rivas. We revised and extended the introduction section and added new citations from the bibliography of Motesharrei et al. (2016). (see the second comment for details on changes in the introduction section).

**REFERENCES**

Lenormand, M., Bassolas, A. and Ramasco, J. J.: Systematic comparison of trip distribution laws and models, J. Transp. Geogr., 51, 158–169, doi:10.1016/J.JTRANGEO.2015.12.008, 2016.

Navarro, A., Moreno, R., Jiménez-Alcázar, A. and Tapiador, F. J.: Coupling population dynamics with earth system models: the POPEM model, Environ. Sci. Pollut. Res., in press, doi:10.1007/s11356-017-0127-7, 2017.

---

## Author Comment (AC5) · 1 Jun 2018

**Response to referee #2**

*Referee #2:*
*General Comments:*
*This paper highlights the importance of grid point scale modeling of anthropogenic pollutants, especially CO2, and the integration of such modeling through a new module called "Population Parameterization for Earth Models" (POPEM). The module is integrated into a highly distributed climate model like Community Earth System Model (CESM). The authors present clearly and adequately the added value of their contribution (POPEM) to the model simulations and underline its impact to the climate predictions of both precipitation and temperature.*

**Reply:** Thanks for your positive feedback.

*Referee #2:*
*Minor comments:*
*Figures 6B and 8B do not illustrate clearly any differences between GPCP – CONTROL, GPCP – POPEM and CRU – CONTROL, CRU – POPEM respectively. Maybe the authors should consider an alternative way to show the differences.*

**Reply:** Thanks for your suggestion. We added new more detailed figures (Figures 9, 11, 12, 13 and 14) to highlight the added value of our approach.

New figures look:

**Figure 9**

[Figure]

*Figure 9: Monthly precipitation (1980-1999) based on GPCP, CTRL and POPEM for three of the regions with important biases in CESM. (A) shows precipitation for the area affected by the double-ITCZ bias in the Southern Hemisphere (20S-0, 80E-100W); (B) for Australia Top End (30S-10S, 128E-140E); and (C) for the Tibetan Plateau (22N-32N, 78W-92W). The black line represents observations (GPCP), the blue line is the CONTROL case, and the red line is the POPEM case. Units are in mm/day. The arrow indicates the improvement of the POPEM model.*

**Figure 11**

[Figure]

***Figure 11****: A comparison of the annual mean surface temperature anomaly between GISTEMP, CONTROL and POPEM from 1950 to 1999. (Top) represents the Barents Sea (68N-80N, 19E-68E); (middle) Russian part of the Bering Sea (50N-65N, 150E-180E); and (bottom) American part of the Bering Sea (50N-75N, 140W-180W). The black line represents observational data (GISTEMP), the blue line is the CONTROL case, and the red is the POPEM case. Anomaly was referenced to 1951-1980 period.*

**Figure 12**

[Figure]

**Figure 12:** A comparison of the global annual mean surface temperature anomaly between GISTEMP, CONTROL, and POPEM from 1950 to 1999. (Top) global; (middle) land; and (bottom) ocean. The black line represents observational data (GISTEMP), the blue line is the CONTROL case, and the red is the POPEM case. Anomaly was referenced to 1951-1980 period.

**Figure 13**

[Figure]

**Figure 13**: Time-series of precipitation anomalies for the ENSO region after Curtis and Adler (2000). (Top) ENSO Precipitation Index (ESPI); (Middle) El Niño Index (EI); and (Bottom) La Niña Index (LI). The Black line shows GPCP data, the blue line is the CONTROL case, and the red line is the POPEM case. Orange shading denotes El Niño years defined as consecutive months (minimum 3) with NIÑO3.4 sea surface temperature anomalies (5N–5S, 170–120W) greater than +0.5$^{\circ}$ C.

**Figure 14**

[Figure]

**Figure 14**: Comparison of the Oceanic el Niño Index (ONI) for CPC (top), POPEM (middle), and CONTROL (bottom) cases. El Niño and La Niña are defined according to Kousky and Higgins (2007): 3-month running mean with anomalies greater than +0.5$^o$C (or-0.5$^o$C) for at least five consecutive months in NIÑO3.4 region. The base period for computing SST departures is 1971–1999.

---

## Author Response (AR2)

**Responses to referee #4**

*Referee #4: I am satisfied with the revision and suggest publication.*

**Reply**: Thanks for your positive feedback.

**Responses to referee #5**

*Referee #5: I would like to express my gratitude to the Authors for addressing the first round of comments from the Reviewers. The manuscript would be ready for publication following the minor changes described below.*

**Reply**: Many thanks for your positive feedback.

*Referee #5: 1. Page 2, Line 10: Change ``Human activity was a major driver of change in the Earth System in the recent past (Alter et al., 2017; Barnett et al., 2008; Crutzen, 2002), and it now dominates the natural system (Ruth, et al. 2011).''*

*To*

*``Human activity has become a major driver of change in the Earth System, especially over the past several decades (Alter et al., 2017; Barnett et al., 2008; Crutzen, 2002), and it now dominates the natural system in many different ways (Motesharrei et al., 2016; Ruth, et al. 2011).''*

**Reply:** We modified the sentence following your suggestion.

*Referee #5: 2. Page 2, Line 26: Delete ``using the predator-prey model''.*

**Reply:** Done.

*Referee #5: 3. Page 5, Line 7: Change ``population dynamics'' to ``population projections''.*

**Reply:** Done.

*Referee #5: 4. Page 6, Line 6: Change ``simulating the dynamics of the population'' to ``simulating the observed population''.*

**Reply:** Done.

*Referee #5: 5. Page 7, Section 2.3: I think it would be helpful to refer to the newly released CESM2, and mention the possibility of coupling POPEM to CESM2 in the future. See, for example:*

*Joel, L. (2018), New version of popular climate model released, Eos, 99, https://doi.org/10.1029/2018EO101489. Published on 22 June 2018.*

**Reply**: Thanks for your suggestion. We added a new line in the future work section.

The text now reads:

> *Future work will be devoted to evaluating the climate response to alternative anthropogenic $CO_2$ emissions; to coupling POPEM with the newest version of CESM (CESM version 2; Joel, L. 2018); to fully coupling Human-Earth subsystems; to increasing the spatial resolution of the simulations; and to refining the spatial and temporal distribution of emission estimates.*

*Referee #5: 6. Page 15, Line 20: Change ``bidirectional feedback'' to ``bidirectional feedbacks''.*

**Reply:** Done.

*Referee #5: 7. Page 15, Line 23: Change ``the counterintuitive feedback that is missing'' to ``the important feedbacks that are missing''.*

**Reply:** Done.

[revised manuscript text omitted]